# Noise Informed LLM for Zero-shot Time Series Forecasting with Uncertainty Quantification

## Abstract

Large language models (LLMs) exhibit strong zero-shot generalization, not only for complex reasoning but also for time-series forecasting. Existing LLM-based forecasters, however, almost exclusively target deterministic accuracy—via elaborate prompts design, tokenization schemes, or instruction tuning—while ignoring the predictive uncertainty that underlies both hallucination and over-confidence. In this work, we bridge this divide by introducing a novel and model-agnostic noise-informed Bayesian approximation (NBA) framework that quantifies the uncertainty of frozen LLMs. We first derive a Bayesian formulation that treats input noise as a stochastic latent variable; marginalizing this noise yields a predictive distribution whose variance is provably calibrated to epistemic plus aleatoric uncertainty. Consequently, the NBA adds negligible overhead, preserves zero-shot accuracy, and avoids the computational cost of posterior inference over LLMs. Systematic experiments on 11 representative LLMs and simulated / real-world datasets show that NBA produces well-calibrated prediction intervals across varying temperature scalings, forecast horizons, model architectures, and prompting strategies. NBA establishes a strong reproducible baseline for uncertainty quantification in LLMs and reveals actionable insights for reliable zero-shot time series forecasting. Code and data are available at
https://anonymous.4open.science/r/NBA-LLM.

## 1 Introduction

The advent of large language models (LLMs) has heralded a paradigm shift in artificial intelligence (AI), demonstrating an unprecedented capacity for zero-shot and few-shot generalization across a diverse spectrum of tasks (Brown et al., 2020). Owing to the efficient information retrieval and representation capabilities of LLMs, they have been widely adopted in fields such as general question answering (QA), finance, healthcare, and education (Chen et al., 2024b; Cheng et al., 2024). Beyond their prowess in natural language generation and understanding, a fascinating and emergent property of these models is their ability to perform complex reasoning in domains far removed from their core training, such as time series (TS) forecasting (Tang et al., 2025; Jin et al., 2023). By leveraging intricate prompt engineering and tokenization mechanisms (Naveed et al., 2024), the application of LLMs to TS forecasting represents an emerging and surprisingly effective paradigm, capitalizing on their innate ability to discern and extrapolate complex temporal patterns in a zero-shot manner. This capability stems from the models' pretraining on vast corpora that implicitly encode sequences, rhythms, and correlations, allowing them to generate forecasts without task-specific fine-tuning (Gruver et al., 2024).

However, it has been observed that LLMs may generate responses that appear plausible but are in fact incorrect or inaccurate, a phenomenon commonly referred to as "hallucination" (Huang et al., 2025; Lin et al., 2022; Li et al., 2025). A significant limitation of this approach lies in its predominant focus on deterministic point predictions, neglecting a cornerstone of trustworthy forecasting: the quantification of predictive uncertainty. Reliable uncertainty quantification (UQ) is indispensable for risk-sensitive decision-making in domains such as finance, epidemiology, and climate science, where understanding the confidence of a forecast is as critical as the forecast itself. Without this, LLMs are prone to overconfident projections or unacknowledged errors, thereby limiting their utility in practical applications. The growing need to quantify predictive uncertainty in high-stakes domains has made it a pressing issue to develop LLMs that can provide reliable UQ.

Broadly, UQ methods can be categorized into two types based on whether they require access to the model's internal parameters: white-box methods and black-box methods. Black-box methods primarily aim to establish correlations between the model's internal output layer and uncertainty, such as CoT-UQ (Zhang & Zhang, 2025), BLoB (Wang et al., 2025). In contrast, white-box methods focus on computing uncertainty values based on multiple responses from the large model, such as semantic entropy (Kuhn et al., 2023) and verbalization (Xiong et al., 2024). However, most of these existing methods concentrate on factual tasks, such as QA and summarization (Fadeeva et al., 2023), where the primary focus is on the correctness of the answers. The estimation of uncertainty in TS forecasting tasks has received relatively limited attention. Current methodologies for UQ in TS are ill-suited for the black-box nature of many contemporary LLMs—particularly closed-source commercial APIs. Furthermore, the computational burden of fine-tuning open-source LLMs is often prohibitive. These constraints collectively necessitate the development of novel black-box UQ (Heo et al., 2025) techniques tailored for temporal reasoning.

In response, we introduce a systematic noise-informed Bayesian approximation (NBA) framework that quantifies the uncertainty of pretrained and frozen LLMs. Given that manipulating the inputs to LLMs is more straightforward than adjusting their parameters, we indirectly apply Bayesian principles to UQ by innovatively estimating the predictive distribution of the outputs conditioned on noisy prompts. Specifically, we introduce noise into the original sequence and treat it as a random variable. By employing Monte Carlo sampling techniques to obtain the predictive likelihood distribution, we can quantify model uncertainty from existing zero-shot black-box LLMs. These noisy TS are tokenized for compatibility with LLMs, and the frozen LLM generates autoregressive forecasts across multiple noise realizations. This framework approximates the predictive distribution via marginalizing over noise: the predictive mean is the average of forecasts from $M$ noise samples, while the variance integrates epistemic (model forecast variability) and aleatoric (inherent noise) uncertainty. Specifically, our contributions are as follows:

- We introduce a novel, model-agnostic Bayesian approximation framework designed to quantify predictive uncertainty in frozen LLMs. This is achieved through injecting carefully calibrated noise into the prompt.

- We establish a rigorous mathematical formulation that provides critical insights into the principles connecting noise-based perturbation to Bayesian marginalization. The derivation of this theoretical foundation not only justifies the use of input noise injection as a valid tool for UQ but also transforms it from a heuristic technique into a well-founded analytical procedure.

- We present an extensive empirical analysis that systematically investigates the influence of critical factors, including temperature scaling, prediction length, model architecture, noise levels, noise distributions, and prompting strategies, on the quality and behavior of the elicited uncertainties. This comprehensive study across diverse datasets and models yields practical insights for implementing UQ in real-world applications and establishes a strong, reproducible baseline for future research in black-box UQ.

## 2 RELATED WORK

**Bayesian Neural Networks (BNNs)**: In statistics and machine learning, uncertainty is modeled in a probabilistic manner. The more dispersed the probability distribution is, the higher the uncertainty appears to be (Hüllermeier & Waegeman, 2021). The Bayesian framework provides a practical tool for uncertainty reasoning in deep learning (Gal & Ghahramani, 2016a). Since the introduction of BNNs (MacKay, 1992), by treating network parameters as random variables with prior distributions, Bayesian deep learning provides a full predictive probability distribution instead of point estimates (Blundell et al., 2015; Xie et al., 2021). However, the sheer size of modern neural networks, with millions or even billions of parameters, makes exact probabilistic inference computationally intractable. Two classes of methods have been proposed to address this. First, sampling techniques like Monte Carlo dropout (Gal & Ghahramani, 2016b), No-U-Turn Sampling (NUTS) (Hoffman et al., 2014), and stochastic gradient MCMC (Welling & Teh, 2011) (Chen et al., 2014) (Zhang et al., 2020) approximate the true posterior distribution by drawing samples from it. Second, approximation methods such as variational inference (Hinton & Van Camp, 1993) (Blundell et al., 2015) (Gal & Ghahramani, 2016a) use a simplified variational distribution to approximate the true posterior,

minimizing the divergence between the two to enable probabilistic predictions. However, recent studies have shown that directly applying the Bayesian framework to LLMs may not be feasible (Lin et al., 2024). This is primarily due to the characteristics of LLMs, which have a large number of internal parameters (Arteaga et al., 2024) and are difficult to train (Xiong et al., 2024), leading to excessive memory and computational costs.

**Uncertainty Quantification in LLMs:** Research in UQ for LLMs is still emerging, especially in NLP (Ling et al., 2024). Some methods rely on internal model information, such as token probabilities (Jiang et al., 2021) or intermediate embeddings (Chen et al., 2024a), which offer robustness but require white-box access and high computational cost. Alternative black-box approaches include prompting models to verbalize numerical confidence (Lin et al., 2022; Xiong et al., 2024), though these are prone to prompt sensitivity and overconfidence (Shorinwa et al., 2024). A notable limitation of such methods is their narrow focus on factual tasks like question answering and summarization, coupled with a lack of mathematical grounding. One line of work explicitly quantifies uncertainty by estimating entropy in the semantic embedding space (Kuhn et al., 2023; Qiu & Miikkulainen, 2024), yet its latent representation must be extracted with an auxiliary deep network, incurring prohibitive computational overhead. Others leverage response consistency as an uncertainty proxy (Wang et al., 2023; Cole et al., 2023; Hou et al., 2024), but these often lack generalizability beyond specific tasks like fact retrieval. Our NBA framework for TS forecasting treats noise as a random variable, eliminating the need for internal access or engineered prompts while achieving good convenience, mathematical rigor, and generalization. In Table 1, we taxonomize UQ methods for LLMs, focusing on QA tasks. The proposed NBA-LLM is uniquely applied to TS forecasting, operating as a mathematically grounded, efficient, black-box Bayesian method without fine-tuning or external tools.

Table 1: A taxonomy of UQ methods for LLMs, categorized by white- or black-box access (W/B), absence of fine-tuning (FT), external tool independence (ET), mathematical grounding (Theo.), computational efficiency (Effi.: low (L) / high (H)), and Bayesian nature (Bayes.).

| Type | Methods | Tasks | W/B | FT | ET | Theo. | Effi. | Bayes. |
|---|---|---|---|---|---|---|---|---|
| Semantic-similarity | (Qiu & Miikkulainen, 2024) | QA | B | ✓ | ✗ | ✓ | L | ✗ |
| | (Ao et al., 2024) | QA | B | ✓ | ✗ | ✓ | L | ✗ |
| | (Kossen et al., 2024) | QA | B | ✓ | ✗ | ✓ | L | ✗ |
| Self-verbalized | (Lin et al., 2022) | QA | B | ✗ | ✓ | ✓ | L | ✗ |
| | (Xiong et al., 2024) | QA | B | ✓ | ✓ | ✗ | H | ✗ |
| | (Band et al., 2024) | QA | B | ✗ | ✓ | ✓ | L | ✗ |
| | (Stengel-Eskin et al., 2024) | QA | B | ✗ | ✗ | ✓ | L | ✗ |
| Latent-information | (Jiang et al., 2021) | QA | W | ✗ | ✓ | ✓ | L | ✗ |
| | (Chen et al., 2024a) | QA | W | ✓ | ✓ | ✓ | H | ✗ |
| | (Ji et al., 2025) | QA | W | ✗ | ✓ | ✓ | L | ✗ |
| Consistency-based | (Manakul et al., 2023) | QA | B | ✓ | ✓ | ✗ | H | ✗ |
| | (Harsha Tanneru et al., 2024) | QA | B | ✓ | ✓ | ✓ | H | ✗ |
| | NBA-LLM (Our) | Time series | B | ✓ | ✓ | ✓ | H | ✓ |

## 3 NOISY PROMPTS AS A BAYESIAN APPROXIMATION

We systematically investigate how to enforce UQ for TS forecasting in LLMs through data perturbation with noise injection and how noisy prompts impact predictive variance.

### 3.1 PROBLEM FORMULATION OF TS FORECASTING

Generally, a TS $\boldsymbol{x} = \{x_t\}_{t=1}^T$ is formally decomposed into a structured signal component $\{f(t)\}_{t=1}^T$ and a stochastic noise component $\{\epsilon_t\}_{t=1}^T$, such that $x_t = f(t) + \epsilon_t$. Here, $f(t)$ captures the underlying temporal dynamics, including trends, cycles, and seasonal patterns. At the same time, $\epsilon_t$ encapsulates irreducible variability and measurement imperfections. The objective of TS forecasting extends beyond point prediction to the probabilistic estimation of future values $\{x_{T+1}, x_{T+2}, \ldots, x_{T+H}\}$ over a horizon $H$, conditioned on historical observations. This is framed as inferring the conditional distribution $p(\{x_t\}_{t=T+1}^{T+H} \mid \{x_t\}_{t=1}^T)$. Within our proposed NBA framework, the noise process is explicitly modeled as an informative random variable, enabling principled UQ and enhanced generalization in a zero-shot learning setting.

## 3.2 UQ OF TS FORECASTING FOR LLM

Formally, UQ of TS forecasting involves inferring a predictive likelihood that marginalizes over both the latent data-generating process and the model parameters (if applicable):

$$p(\mathbf{x}_{T+1:T+H} \mid \mathbf{x}_{1:T}) = \int p(\mathbf{x}_{T+1:T+H} \mid \boldsymbol{\theta}, \mathbf{x}_{1:T}) p(\boldsymbol{\theta} \mid \mathbf{x}_{1:T}) d\boldsymbol{\theta}, \tag{1}$$

where $\boldsymbol{\theta}$ represents the model parameters or latent variables. In cases where the model is treated as a black box (e.g., a pretrained LLM) and parameter uncertainty is not directly accessible, UQ must be performed through alternative strategies. When leveraging LLMs for TS forecasting, the series is often tokenized into symbolic sequences $\mathbf{s}_{1:n}$, and forecasting becomes an autoregressive sequence generation task. The UQ objective thus translates to quantifying uncertainty in this token-level generative process, accounting for both the variability in token predictions and the propagation of uncertainty through sequential steps. We suppose that a robust UQ method should therefore: 1) provide well-calibrated probabilistic forecasts, 2) remain computationally tractable without requiring internal model modifications, and 3) generalize across varying forecast horizons and model architectures.

## 3.3 BAYESIAN MARGINALIZATION

The core challenge for TS forecasting in LLMs lies in quantifying the predictive uncertainty of the LLM without modifying its parameters or incurring significant computational overhead. Because the parameter uncertainty of LLMs is precluded, Eq. 1 underscores the need for alternative approaches such as Bayesian approximation, noise injection, or sampling strategies that yield a distribution over plausible futures rather than a single deterministic trajectory. Therefore, we define a mathematically grounded and efficient Bayesian marginalization in the NBA framework that treats the LLM as a black-box function $f(t)$ subject to input perturbations. Let $H = 1$ and $\delta \sim p(\delta)$ be a noise variable injected into the TS or its embedding. The predictive distribution is approximated via marginalization over this noise:

$$
\begin{aligned}
p(\mathbf{x}_{T+1} \mid \mathbf{x}_{1:T}) &= \int p(\mathbf{x}_{T+1} \mid \boldsymbol{\delta}, \mathbf{x}_{1:T}) p(\boldsymbol{\delta} \mid \mathbf{x}_{1:T}) d\boldsymbol{\delta}, \\
&= \int p(\hat{f}(\mathbf{x}_{1:T}, \boldsymbol{\delta})) p(\boldsymbol{\delta}) d\boldsymbol{\delta},
\end{aligned}
\tag{2}
$$

where $f$ denotes the deterministic forward pass of the frozen LLM. From a probabilistic perspective, the target predictive distribution is formulated as a Bayesian model average. Rather than relying on a single deterministic forward pass of the LLM $f(x)$, the NBA framework incorporates multiple realizations of the input noise $\delta$, each weighted by its probability. This marginalization over $\delta$ follows directly from the sum and product rules of probability, allowing the model to account for predictive uncertainty without modifying the underlying LLM parameters. By treating noise as a key source of uncertainty, the approach facilitates robust probabilistic forecasting in a zero-shot setting.

## 3.4 OBTAINING MODEL UNCERTAINTY VIA BAYESIAN APPROXIMATION

Building upon the Bayesian marginalization, we demonstrate that model uncertainty can be effectively quantified. Due to the intractable integral in Eq. 2, we employ moment-matching to estimate the first two moments of the distribution empirically.

**Proposition 1** *Given the predictive distribution $p(\mathbf{x}_{T+1} \mid \mathbf{x}_{1:T})$, the corresponding predictive mean admits the Monte Carlo approximation*

$$
\begin{aligned}
\mathbb{E}_{p(\mathbf{x}_{T+1} \mid \mathbf{x}_{1:T})}(\mathbf{x}_{T+1}) &= \int \hat{f}_{T+1}(\mathbf{x}_{1:T}, \boldsymbol{\delta}) \, p(\boldsymbol{\delta}) \, d\boldsymbol{\delta}, \\
&\approx \frac{1}{M} \sum_{m=1}^{M} \hat{f}_{T+1}(\mathbf{x}_{1:T}, \boldsymbol{\delta}_m), \quad \boldsymbol{\delta}_m \sim p(\delta).
\end{aligned}
\tag{3}
$$

*where $\hat{f}(\mathbf{x}_{1:T}, \boldsymbol{\delta})$ denotes the LLM forecast under noise realization $\delta \sim p(\delta)$, $M$ is the number of independent noise realizations.*

**Proposition 2** *The predictive variance of the future value $\mathbf{x}_{T+1}$ under the NBA framework can be approximated via Monte Carlo sampling as:*

$$\mathrm{Var}_{p(\mathbf{x}_{T+1}|\mathbf{x}_{1:T})}(\mathbf{x}_{T+1}) = \mathbb{E}_{p(\delta)}[\sigma_*^2] + \mathrm{Var}_{p(\delta)}[\hat{f}_{T+1}(\mathbf{x}_{1:T}, \boldsymbol{\delta}_m)],$$

$$\approx \frac{1}{M}\sum_{m=1}^{M}\hat{f}_{T+1}(\mathbf{x}_{1:T}, \boldsymbol{\delta}_m)^2 - \left(\frac{1}{M}\sum_{m=1}^{M}\hat{f}_{T+1}(\mathbf{x}_{1:T}, \boldsymbol{\delta}_m)\right)^2 + \sigma_\delta^2, \tag{4}$$

*where $\sigma_*^2$ is the variance of the predictive distribution $p(\mathbf{x}_{T+1} \mid \mathbf{x}_{1:T}, \boldsymbol{\delta})$ for a given noise $\boldsymbol{\delta}$, $\sigma_\delta^2$ denotes the noise variance.*

Hence, we derived and proved that a mathematically grounded model uncertainty estimate can be obtained from LLMs with a prompt-noising strategy. The detailed derivation process is provided in Appendix A.

**Noise Design and Sampling Strategies**. In the context of the NBA, we specify a tractable prior distribution for the noise variable, typically Gaussian, denoted as $p(\delta)$. From a predictive estimation standpoint, this Monte Carlo procedure approximates the predictive likelihood using discrete point masses situated at samples drawn from the continuous prior, such that $p(\delta) \approx \sum_{m=1}^{M} \delta(\delta = \delta_m)$ for $\delta_m \sim p(\delta)$. The injected noise is modeled as a random variable with zero mean and variance $\sigma_\delta^2$, where the variance quantifies the uncertainty inherent in the observational process. Gaussian noise, for instance, is expressed as $\delta_m \sim \mathcal{N}(0, \sigma_\delta^2)$. Beyond Gaussian assumptions, we also investigate uniform, Laplace, Gamma, and Beta distributions to assess robustness under various noise structures. To regulate the influence of noise relative to the underlying signal, we incorporate a scaling mechanism that adjusts the noise magnitude in a controlled manner. This is formalized by parameterizing the noise variance as $\sigma_\delta^2 = \alpha^2 \sigma_x^2$, where $\sigma_x^2$ is the variance of the original TS and $\alpha$ is a scaling factor that modulates the noise intensity. This approach ensures that the injected noise meaningfully influences model behavior without dominating the true signal, thereby balancing sensitivity and robustness in the forecasting process. The resulting noise amplitude is thus jointly determined by the data variability $\sigma_x$ and the tunable scaling factor $\alpha$.

### 3.5 TOKEN MODELING AND PREDICTION IN LLM

**Tokenization of Noisy TS**. Within the NBA framework, noise injection is formalized as a stochastic perturbation operator $\mathcal{P} : \mathbb{R} \to \mathbb{R}$ defined by

$$\mathcal{P}(\mathbf{x}_t) = \tilde{\mathbf{x}}_t = \mathbf{x}_t + \delta_t, \tag{5}$$

where $\delta_t$ is sampled from a noise distribution with $\mathbb{E}[\delta_t] = 0$. This perturbation encourages the model to prioritize robust latent temporal structures over incidental fluctuations, thereby enhancing generalization without architectural changes or retraining. The perturbed series $\{\tilde{\mathbf{x}}_t\}$ is then processed by the LLM, improving robustness to distributional shifts and enabling uncertainty-aware forecasting. To interface numerical TS with transformer-based LLMs, a tokenization operator $\mathcal{Q} : \mathbb{R}^T \to \mathcal{S}^T$ bijectively maps the noised series $\{\tilde{\mathbf{x}}_t\}_{t=1}^{T}$ into a discrete token sequence $S = \{\mathrm{Token}_t(\tilde{\mathbf{x}}_t)\}_{t=1}^{T}$, where $\mathcal{S}$ denotes the token vocabulary. This mapping preserves invertibility, satisfying

$$S = \mathcal{Q}(\{\tilde{\mathbf{x}}_t\}) \quad \text{and} \quad \{\tilde{\mathbf{x}}_t\} = \mathcal{Q}^{-1}(S), \tag{6}$$

ensuring faithful representation between numerical inputs and symbolic sequences. This tokenization enables the LLM to leverage its sequence modeling capabilities for zero-shot forecasting, while the injected noise provides a mechanism for Bayesian UQ through stochastic forward passes.

**Token Prediction**. LLMs are trained on sequential data $\mathcal{S} = \{S_1, S_2, \ldots, S_N\}$, where each sequence $S_i$ consists of tokens from a vocabulary $\mathcal{V}$. These models learn an autoregressive distribution $p_\Theta(S_i) = \prod_{j=1}^{n_i} p_\Theta(\mathbf{s}_{i,j} \mid \mathbf{s}_{i,0:j-1})$, with parameters $\Theta$ optimized to maximize the corpus likelihood $p_\Theta(\mathcal{S}) = \prod_{i=1}^{N} p_\Theta(S_i)$. In the NBA framework, TS data are treated as token streams, allowing the LLM to capture implicit dynamics. By integrating Bayesian principles, the model facilitates UQ without retraining. In this context, token prediction initiates from a noisy prompt sequence $\mathbf{s}_{0:k}$ and proceeds autoregressively according to the distribution $p_\Theta(\mathbf{s}_j \mid \mathbf{s}_{0:j-1})$. Within this formulation, TS forecasting is reframed as a conditional sequence generation problem. The autoregressive predictive distribution for a future time point is expressed as $p(\mathrm{Token}(\mathbf{x}_{T+1}) \mid \{\mathrm{Token}(\mathbf{x}_t)\}_{t=1}^{T})$, thereby

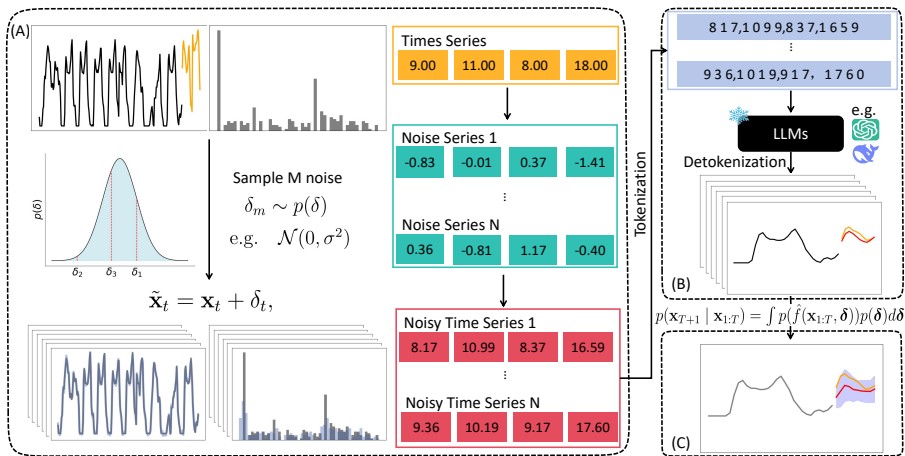

Figure 1: Pipeline of NBA-LLM: a lightweight and model-agnostic Bayesian-LLM with UQ for zero-shot TS forecasting. Box (A): Monte Carlo sampling of TS with noise injection. Box (B): Token prediction of frozen LLM with noisy prompt. Box (C): zero-shot TS forecasting with UQ.

enabling the approximation of the forecast distribution $p(\tilde{\mathbf{x}}_{T+1} \mid \{\tilde{\mathbf{x}}_t\}_{t=1}^T)$ through token-level predictive probabilities. Consequently, the predictive distribution is approximated as

$$p(\mathbf{x}_{T+1}|\{\tilde{\mathbf{x}}_t\}_{t=1}^T) \approx p(\text{Token}(\mathbf{x}_{T+1})|\{\text{Token}(\tilde{\mathbf{x}}_t)\}_{t=1}^T). \tag{7}$$

## 3.6 Framework of NBA

In Fig. 1, we present the procedural pipeline of the NBA-LLM for zero-shot TS forecasting with integrated UQ. In the Box (a), the gray curve represents the training set of the true sequence, the orange curve represents the test set of the true sequence, and the blue curve represents the training set with added noise. At each time point, the noise is completely random, causing fluctuations in the data, either increasing or decreasing. However, overall, the distribution shape of the original sequence and the perturbed sequence is approximately similar. This indicates that our perturbed sequence still retains sufficient original structural features, successfully simulating the uncertainty of the data. In the bar chart, gray represents the true sequence, and blue represents the perturbed sequence. In the Box (B), the predicted sequence (in red line) can fluctuate in accordance with the true sequence, but there is still a certain deviation from the true sequence. This highlights the importance of evaluating the uncertainty of LLMs in TS forecasting tasks. By quantifying the uncertainty of the large language model's predictions, we aim to reflect its confidence in the prediction results. In the Box (C), we observe that even though there is a certain gap between the predicted sequence and the true sequence, the confidence interval (in the blue area) still manages to cover the original sequence.

## 4 Experiments

To rigorously evaluate the efficacy of the NBA-LLM framework, we conduct an extensive empirical study for zero-shot TS forecasting and UQ. The experiments are structured to systematically investigate the impact of various critical factors on the quality of the predictive distribution and the calibration of UQ. We consider a series of benchmark datasets, including Darts (Herzen et al., 2022), Informer (Zhou et al., 2021), and Memorization (Gruver et al., 2024). Detailed experiments are provided in the Appendix C.

**Model.** To ensure a representative evaluation of NBA-LLM, we select a diverse set of LLMs spanning multiple architectural families and scaling regimes. The evaluated models include the GPT series (OpenAI et al., 2024), Claude models (Team et al., 2024), GLM-4 (GLM et al., 2024), Gemini Flash 2.0, Qwen series (Qwen et al., 2025), and DeepSeek models (Zhang et al., 2025). This spectrum covers both instruction-tuned (IT) and reasoning specialized variants. However, some of the latest or more complex LLMs were not included, primarily due to cost considerations.

**Metrics.** We evaluate UQ using the negative log-likelihood (NLL), which measures sharpness at the true value, and the continuous ranked probability score (CRPS), which assesses overall distributional

calibration. These metrics offer a rigorous probabilistic benchmark. In addition, the Normalized Mean Squared Error (NMSE) is employed to complement probabilistic metrics by quantifying the precision of the predictive mean. Direct numerical comparisons with other methods are avoided, as results under differing protocols are not statistically comparable.

## 4.1 UQ OF LLMs ON SYNTHETIC DATA

To validate the efficacy of the NBA-LLM for zero-shot TS forecasting with UQ, we generate synthetic data by sampling from a Gaussian process (GP). The use of synthetic data eliminates the risk of data leakage. This guarantees that the LLM, operating in a strict zero-shot regime, has had absolutely no prior exposure—direct or indirect—to the test sequences. A series of 300 points is sampled from the GP, with added observational noise introduced to 20% of the points to simulate real-world data imperfections. The series is partitioned into 270 points for context and 30 points for testing.

As shown in Fig. 2, the synthetic TS superimposes a smooth trend with abrupt, noise-driven irregularities, deliberately increasing the difficulty of uncertainty estimation. The NBA-LLM predictions not only accurately track the underlying trend but also produce well-calibrated uncertainty (indicated by confidence interval (CI)) that closely envelops the ground-truth values. Importantly, the predictive intervals exhibit in-

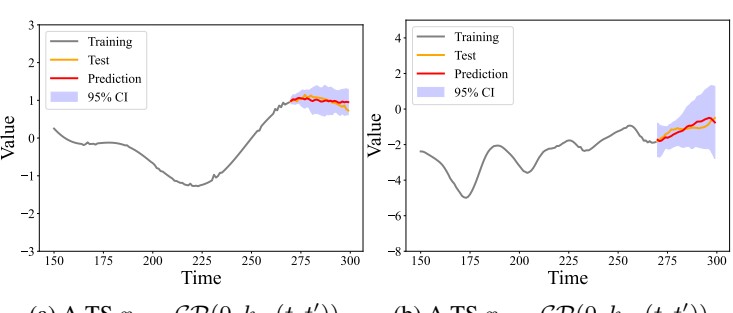

(a) A TS $x_t \sim \mathcal{GP}(0, k_{\mathrm{SE}}(t, t'))$  (b) A TS $x_t \sim \mathcal{GP}(0, k_{\mathrm{RQ}}(t, t'))$

Figure 2: UQ of NBA-LLM (GPT-3.5-Turbo model) on synthetic TS sampled from GPs with squared exponential (SE) kernel $k_{\mathrm{SE}}(t, t')$ and rational quadratic (RQ) kernel $k_{\mathrm{RQ}}(t, t')$.

creasing width with forecasting horizon, reflecting the accumulation of uncertainty over time—a key characteristic of principled probabilistic forecasting. This result underscores the NBA's capacity for robust UQ without task-specific training.

## 4.2 NBA-LLM WITH UQ FOR REAL-WORLD TS FORECASTING

As shown in Table 2, we applied NBA across various LLMs to gain preliminary insights into their performance and UQ in TS forecasting. More results are detailed in Appendices C.9, C.10 and C.11. Among closed-source models, GPT-4 demonstrates superior performance, while GLM-4 emerges as the leading open-source alternative. However, both approaches exhibit substantial performance gaps when compared to GPT-4. Although closed-source models achieve significantly higher accuracy, our analysis reveals that uncertainty estimation capabilities remain comparable between closed-source and open-source paradigms, with neither demonstrating clear advantages in uncertainty calibration. Notably, we observe anomalous behaviors in specific model-dataset configurations: GPT-3.5-Turbo-Instruct and Gemini Flash 2.0 (lite) exhibit exceptionally high NLL values on particular datasets, indicating outlier peaks in predicted probability densities. This phenomenon suggests inherent model overconfidence. Surprisingly, DeepSeek-R1, despite its renowned reasoning capabilities, demonstrates uncontrolled uncertainty propagation in temporal tasks. This unexpected degradation may stem from alignment interventions, particularly Reinforcement Learning from Human Feedback (RLHF), which appears to introduce unintended side effects in UQ for TS applications. Our findings underscore that UQ in zero-shot TS forecasting remains a formidable challenge for current LLMs.

## 4.3 COMPARATIVE BAYESIAN MARGINALIZATION: TEMPERATURE VS. NOISE

Table 2: UQ of LLMs on the Memorize, Darts, and Informer datasets. The models are abbreviated as follows: Clau. $3.5H$ (Claude-3.5-Haiku), Clau. $3.5S$ (Claude-3.5-Sonnet), QW (Qwen), and DS (DeepSeek). Gemini refers to Gemini Flash 2.0. Subscripts T and I indicate Turbo and Instruct models, respectively, and the superscript R denotes a reasoning model.

| Model | NMSE | | | CRPS | | | NLL | | |
|---|---|---|---|---|---|---|---|---|---|
| | Memorization | Darts | Informer | Memorization | Darts | Informer | Memorization | Darts | Informer |
| **Closed-source LLM** | | | | | | | | | |
| GPT-3.5$_T$ | 1.50±0.37 | 1.43±0.26 | 2.32±0.26 | 0.14±0.08 | 0.15±0.06 | 0.28±0.04 | 7.76±1.76 | **6.22**±0.91 | **3.56**±0.65 |
| GPT-3.5$_{IT}$ | 1.07±0.68 | 1.26±0.22 | 3.88±0.79 | 0.14±0.08 | 0.16±0.07 | 0.46±0.08 | 6.17±1.16 | 6.56±0.63 | 966.78±423.86 |
| GPT-4 | **0.81**±0.28 | **0.81**±0.18 | 2.05±0.20 | 0.13±0.07 | **0.14**±0.06 | 0.28±0.04 | **6.03**±1.24 | 6.66±0.90 | 6.72±1.48 |
| Clau. $3.5_H$ | 1.36±0.50 | 1.73±0.36 | 2.55±0.36 | 0.15±0.09 | 0.17±0.06 | **0.26**±0.03 | 6.87±1.88 | 13.09±5.07 | 15.09±3.67 |
| Clau. $3.5_S$ | 4.21±2.14 | 1.42±0.41 | 5.67±1.01 | **0.13**±0.06 | 0.16±0.06 | 0.33±0.06 | 90.44±64.96 | 9.46±2.05 | 26.71±14.02 |
| **Average** | 1.79 | 1.33 | 3.29 | 0.14 | 0.16 | 0.32 | 23.45 | 8.40 | 203.77 |
| **Open-source LLM** | | | | | | | | | |
| GLM-4 | **1.30**±0.51 | **1.52**±0.24 | **2.23**±0.23 | 0.18±0.11 | **0.17**±0.06 | 0.27±0.04 | **6.30**±1.45 | **6.48**±0.95 | **3.63**±0.65 |
| Gemini | 2.42±1.53 | 14.14±5.03 | 2.78±0.51 | 0.23±0.15 | 0.19±0.07 | **0.26**±0.03 | 7.05±1.37 | 1013.08±910.75 | 6.14±1.45 |
| QW$_T$ | 1.53±0.49 | 2.14±0.41 | 2.98±0.43 | **0.12**±0.07 | 0.22±0.10 | 0.31±0.05 | 8.48±1.81 | 9.21±1.94 | 24.34±7.14 |
| QW2.5$_{IT}$ | 2.20±0.43 | 8.32±4.72 | 2.90±0.46 | 0.16±0.10 | 0.18±0.08 | 0.29±0.04 | 8.57±1.92 | 169.22±149.76 | 17.70±4.34 |
| DS-R1 | 2.78±1.18 | 1.78±0.37 | 3.83±0.62 | 0.27±0.19 | 0.18±0.06 | 0.38±0.05 | 14.88±2.42 | 15.72±4.26 | 145.61±52.01 |
| DS-V3 | 1.84±0.90 | 2.25±0.73 | 5.65±1.10 | 0.17±0.11 | 0.17±0.06 | 0.35±0.06 | 6.59±1.22 | 7.15±0.74 | 27.47±15.37 |
| **Average** | 2.20 | 5.03 | 3.43 | 0.19 | 0.18 | 0.31 | 8.51 | 1477.96 | 35.03 |

The temperature parameter in LLMs is a scaling factor applied to the logits prior to the softmax operation in the final output layer, formally defined as $P(\text{Token}) = \text{softmax}(\text{logits}/\tau)$, where $\tau$ denotes the temperature and $P(\text{Token})$ is the corresponding probability. The uncertainty in the noise injection strategy is primarily introduced by altering the data, whereas the uncertainty in the temperature scaling strategy is mainly introduced by controlling the entropy of the resulting probability distribution over the vocabulary. By treating the temperature as the

Table 3: UQ of NBA-LLM with temperature and noise marginalization on the Memorize dataset.

| Time series | Temperature | | | Noise | | |
|---|---|---|---|---|---|---|
| | NMSE | CRPS | NLL | NMSE | CRPS | NLL |
| IstanbulTraffic | 2.35 | 0.31 | 8.82 | 2.36 | 0.33 | 8.06 |
| TSMCStock | 2.23 | 0.02 | 4.48 | 0.80 | 0.02 | 3.89 |
| TurkeyPower | 1.56 | 0.06 | 24.80 | 1.34 | 0.06 | 11.33 |

latent variable to be integral in the Bayesian marginalization, we have the formula for the temperature strategy as $p(\mathbf{x}_{T+1} \mid \mathbf{x}_{1:T}) = \int p(\hat{f}(\mathbf{x}_{1:T}, \tau)) p(\tau) d\tau$. As shown in Table 3, noise injection uniformly outperforms temperature scaling in NBA-LLM, with the gap most pronounced on NLL, indicating that the former yields better-calibrated and more trustworthy UQ. This result cautions that aggressive temperature tuning can seed low-probability outliers during autoregressive generation; consequently, careful temperature initialization should be treated as a first-class design decision rather than a post-hoc afterthought. Furthermore, we visualize the UQ under the temperature-scaling strategy in Fig. 3.

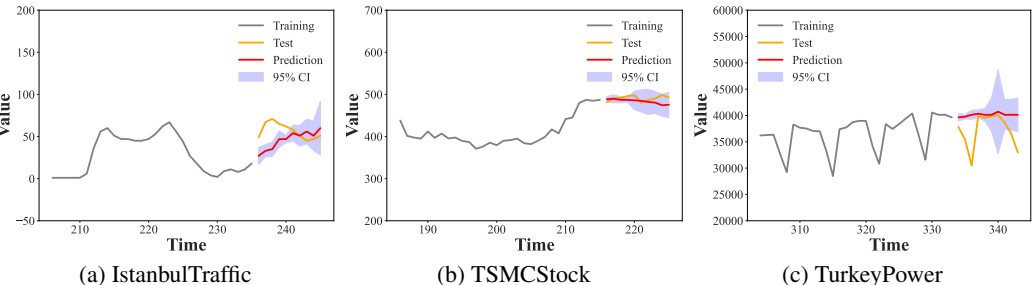

(a) IstanbulTraffic    (b) TSMCStock    (c) TurkeyPower

Figure 3: Uncertainty-aware prediction of NBA-LLM (GPT-3.5-turbo) with temperature marginalization on the Memorization dataset.

## 4.4 ABLATION STUDY

In this section, we meticulously analyze the influence of a comprehensive set of parameters, including forecast horizon, noise levels, noise distribution specifications, sampling temperature, model scale, prompt engineering strategies, and underlying model architecture.

**Forecast Horizon Proportionally Inflates Uncertainty**. In the field of TS forecasting, traditional machine learning methods often categorize tasks based on their prediction horizon, namely, short-term versus long-term forecasting. This study systematically evaluates the performance of GPT-3.5-Turbo across two distinct prediction horizons—96 and 192 steps—to offer a more comprehensive perspective. As shown in Fig. 6, NBA-LLMs consistently produce stable and reasonable results for both short- and long-term horizons. This finding indicates a promising path forward for extending LLMs to achieve highly effective long-term forecasting with UQ.

**Sweet-Spot Noise Improves Calibration and Excess Noise Destroys It.** In NBA-LLM, the noise level $\sigma_\delta$ directly controls the noise variance. Theoretically, as the injected variance increases, so does the apparent volatility of the series, monotonically amplifying the complexity of reliable UQ. In Fig. 8, we plot the LLM estimation metrics for the Darts dataset under varying noise levels. Across all noise levels, the overall metrics for the Darts collection remain relatively constant, showing only mild fluctuations. The general trend is a slight decrease followed by an increase. This suggests that injecting low levels of noise during inference can be considered an effective UQ technique.

**Noise Following Heavy-Tailed Gamma Distribution Yields Better Calibration.** Beyond the noise levels, the distribution of noise also influences the distribution of input data, thereby affecting the performance of NBA-LLM. We primarily introduced six types of noise distributions: Gaussian, uniform, geometric, Laplace, Gamma, and Beta. The specific forms of these distributions are detailed in Appendix C.5. As shown in Fig. 9, under all noise-injection conditions, the TS predictions closely track the fluctuations of the true values, and the confidence intervals encompass the majority of the true values, demonstrating that the NBA-LLM method exhibits good generalizability and robustness to different noise distributions. Note that noise sampled from a heavy-tailed Gamma distribution yields superior calibration properties. This is attributed to the distribution's capacity to generate more diverse and extreme perturbations, which better explores the function space of the LLM during the Monte Carlo marginalization process.

**Temperature Scaling Induces Minor Changes in Calibration.** We conduct a systematic evaluation of GPT-3.5-Turbo on the Memorization dataset, sweeping temperature $\tau \in [0, 2]$. As shown in Fig. 10, all three metrics exhibit minimal variance across the entire range, confirming that the model's TS forecasts are remarkably robust to temperature rescaling. Notably, no monotonic trend emerges; instead, intermediate temperatures ($\tau \approx [0.8, 1.2]$) consistently occupy a broad, low-error plateau, making this interval a safe default when calibration stability is paramount.

**Text-First Prompts Undermine UQ in TS Forecasting.** To investigate the effect of specific prompts on LLM-Time's forecasting uncertainty, we tested several common human-heuristic prompting strategies in this section. These strategies have been repeatedly shown to significantly influence model output in commonsense question-answering tasks, including: Direct, CoT (Wei et al., 2023; Kojima et al., 2023), Self-Probing (Baek et al., 2025), Self-Correcting (Kim et al., 2023; Madaan et al., 2023; Kumar et al., 2024), Prompt-Optimizer (Shen, 2025). For the full prompt, refer to Appendix D.2. Surprisingly, Table 6 reveals that text-first prompts impair both predictive accuracy and UQ on numerical tasks. Augmenting the prompt with additional cognitive stages (e.g., explicit reasoning or self-correction) systematically degrades performance.

## 5 CONCLUSION

In this work, we focus on quantifying the uncertainty of LLMs in TS forecasting tasks using Bayesian methods. Specifically, we introduce noise into the original sequence and treat it as a random variable. By employing Monte Carlo sampling techniques to obtain the predictive likelihood distribution of predictions, we can quantify model uncertainty from existing zero-shot black-box LLMs. This approach not only eliminates the need to access the internal parameters of large models but is also applicable to both open-source and closed-source models. It significantly reduces computational resources and does not require the careful design of prompts. As a zero-shot prediction task, it dramatically lowers the technical threshold, demonstrating strong versatility, convenience, and cost-effectiveness. We conducted extensive benchmarking using LLMs on a synthetic dataset and three real-world datasets. Our results show that the noise injection strategy consistently enhances predictive performance and provides reasonable UQ across all datasets, outperforming Bayesian methods based on temperature strategies. Moreover, we performed a comprehensive set of ablation studies, analyzing and conducting sensitivity analyses on eight factors: short-term and long-term predictions (data level),

noise levels and noise distributions (noise level), temperature parameters, model sizes, prompt styles, sample sizes, and model types (model level).

## ETHICS STATEMENT

This work advances the development of safer AI systems by providing calibrated probabilistic forecasts, crucial for high-stakes domains like finance, where overreliance on deterministic predictions poses significant risks. While our framework enhances uncertainty quantification, responsible deployment requires context-specific validation to ensure proper interpretation and action based on the uncertainty estimates.

## REPRODUCIBILITY STATEMENT

We release a fully open-source, zero-shot pipeline that turns off-the-shelf LLMs into principled uncertainty quantification for TS forecasting. The workflow requires neither fine-tuning nor task-specific training—only lightweight API calls—eliminating dependence on proprietary architectures or expensive retraining. By lowering these practical barriers, we aim to accelerate community-wide progress on reliable, large-scale generative modeling. Source code, complete proofs, and experimental datasets are provided under the MIT licence in the Appendix.

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

# A    PROPOSITION AND PROOF

**Proposition 3** *The predictive log-likelihood for the future value* $\mathbf{x}_{T+1}$ *under the NBA framework can be approximated via Monte Carlo sampling as:*

$$\log p(\mathbf{x}_{T+1} \mid \mathbf{x}_{1:T}) = \log \int p(\mathbf{x}_{T+1} \mid \mathbf{x}_{1:T}, \boldsymbol{\delta}) p(\boldsymbol{\delta}) d\boldsymbol{\delta}$$

$$\approx \log \Big( \frac{1}{M} \sum_{m=1}^{M} p(\mathbf{x}_{T+1} \mid \mathbf{x}_{1:T}, \boldsymbol{\delta}_m) \Big). \tag{8}$$

*Assuming a Gaussian observation model* $p(\mathbf{x}_{T+1} \mid \mathbf{x}_{1:T}, \boldsymbol{\delta}) = \mathcal{N}(\mathbf{x}_{T+1}; \hat{f}_{T+1}(\mathbf{x}_{1:T}, \boldsymbol{\delta}), \sigma_\epsilon^2)$, *this simplifies to:*

$$\log p(\mathbf{x}_{T+1} \mid \mathbf{x}_{1:T}) \approx \text{logsumexp}_{m=1}^{M} \Big( - \frac{(\mathbf{x}_{T+1} - \hat{f}_{T+1}(\mathbf{x}_{1:T}, \boldsymbol{\delta}_m))^2}{2\sigma_\delta^2} \Big) - \log M - \frac{1}{2} \log(2\pi\sigma_\epsilon^2),$$

*where* logsumexp *denotes the log-sum-exp operator.*

## A.1    PROOF OF PREDICTIVE MEAN

Starting from the definition of the predictive distribution:

$$p(x_{T+1} \mid x_{1:T}) = \int p(x_{T+1} \mid x_{1:T}, \delta) \, p(\delta) \, d\delta.$$

The expectation of $x_{T+1}$ is therefore:

$$\mathbb{E}_{p(x_{T+1} \mid x_{1:T})}(x_{T+1}) = \int x_{T+1} \, p(x_{T+1} \mid x_{1:T}) \, dx_{T+1} = \iint x_{T+1} \, p(x_{T+1} \mid x_{1:T}, \delta) \, p(\delta) \, d\delta \, dx_{T+1}.$$

Exchanging the order of integration and recognizing that the inner integral yields the model's forecast $\hat{x}_{T+1}(x_{1:T}, \delta)$, we obtain:

$$\mathbb{E}_{p(x_{T+1} \mid x_{1:T})}(x_{T+1}) = \int \hat{x}_{T+1}(x_{1:T}, \delta) \, p(\delta) \, d\delta.$$

The Monte Carlo estimate follows directly from this integral representation.

## A.2    PROOF OF PREDICTIVE VARIANCE

The predictive variance is defined as:

$$\text{Var}_{p(\mathbf{x}_{T+1} \mid \mathbf{x}_{1:T})}(\mathbf{x}_{T+1}) = \mathbb{E}_{p(\mathbf{x}_{T+1} \mid \mathbf{x}_{1:T})}[\mathbf{x}_{T+1}^2] - \big(\mathbb{E}_{p(\mathbf{x}_{T+1} \mid \mathbf{x}_{1:T})}[\mathbf{x}_{T+1}]\big)^2.$$

We begin by expressing the second raw moment via the law of total expectation:

$$\mathbb{E}[\mathbf{x}_{T+1}^2 \mid \mathbf{x}_{1:T}] = \mathbb{E}_{p(\delta)} \big[\mathbb{E}[\mathbf{x}_{T+1}^2 \mid \mathbf{x}_{1:T}, \delta]\big].$$

For a fixed $\delta$, the inner expectation decomposes as:

$$\mathbb{E}[\mathbf{x}_{T+1}^2 \mid \mathbf{x}_{1:T}, \delta] = \text{Var}[\mathbf{x}_{T+1} \mid \mathbf{x}_{1:T}, \delta] + (\mathbb{E}[\mathbf{x}_{T+1} \mid \mathbf{x}_{1:T}, \delta])^2 = \sigma_x^2 + \hat{f}_{T+1}(\mathbf{x}_{1:T}, \delta)^2.$$

Substituting back, the second moment becomes:

$$\mathbb{E}[\mathbf{x}_{T+1}^2 \mid \mathbf{x}_{1:T}] = \mathbb{E}_{p(\delta)}[\sigma_x^2 + \hat{f}_{T+1}^2] = \mathbb{E}_{p(\delta)}[\sigma_x^2] + \mathbb{E}_{p(\delta)}[\hat{f}_{T+1}^2].$$

From Proposition 3, the first moment is $\mathbb{E}[\mathbf{x}_{T+1} \mid \mathbf{x}_{1:T}] = \mathbb{E}_{p(\delta)}[\hat{f}_{T+1}]$. Therefore, the predictive variance is:

$$\text{Var}[\mathbf{x}_{T+1} \mid \mathbf{x}_{1:T}] = \Big(\mathbb{E}_{p(\delta)}[\sigma_x^2] + \mathbb{E}_{p(\delta)}[\hat{f}_{T+1}^2]\Big) - \Big(\mathbb{E}_{p(\delta)}[\hat{f}_{T+1}]\Big)^2.$$

Recognizing that $\mathbb{E}_{p(\delta)}[\hat{f}_{T+1}^2] - \Big(\mathbb{E}_{p(\delta)}[\hat{f}_{T+1}]\Big)^2 = \text{Var}_{p(\delta)}[\hat{f}_{T+1}]$, we obtain the final expression:

$$\text{Var}[\mathbf{x}_{T+1} \mid \mathbf{x}_{1:T}] = \mathbb{E}_{p(\delta)}[\sigma_x^2] + \text{Var}_{p(\delta)}[\hat{f}_{T+1}].$$

The Monte Carlo approximation follows directly by estimating each term with samples $\delta_m \sim p(\delta)$:

$$\text{Var}[\mathbf{x}_{T+1} \mid \mathbf{x}_{1:T}] \approx \frac{1}{M} \sum_{m=1}^{M} \sigma_{\delta_m}^2 + \left( \frac{1}{M} \sum_{m=1}^{M} \hat{f}_{T+1}^2 - \left( \frac{1}{M} \sum_{m=1}^{M} \hat{f}_{T+1} \right)^2 \right).$$

## B   ALGORITHM OF NBA

In Algorithm. 1, the methodology commences with a Monte Carlo noise injection stage, wherein the original observed sequence $\mathbf{x}_{1:T}$ is perturbed by $M$ independent noise realizations $\delta_m$ drawn from a prescribed distribution, such as $\mathcal{N}(0, \sigma^2)$. This operation produces $M$ noised variants of the input, formally expressed as Eq. 5 , thereby constructing an ensemble of plausible input scenarios that embody aleatoric uncertainty at the data level. Each perturbed series $\tilde{\mathbf{x}}_{1:T}$ is subsequently mapped into a discrete token sequence via a deterministic tokenization operator, rendering it suitable for processing by a frozen LLM. The LLM executes an autoregressive forward pass on each tokenized sequence, generating a corresponding predictive distribution over subsequent values, symbolically represented as $p(\text{Token}(\mathbf{x}_{T+1}) \mid \{\text{Token}(\mathbf{x}_t)\}_{t=1}^{T})$. This step effectively propagates input-level stochasticity through the model, inducing functional diversity in the forecasts without any internal parameter adjustments. The final phase involves statistical aggregation of the $M$ independent predictive outputs to approximate the predictive likelihood. The predictive mean is estimated as Eq. 3, while the total predictive variance is derived from Eq. 4 across the ensemble, capturing both epistemic uncertainty (via the variance of the forecasts) and aleatoric uncertainty (via the average internal variance of each prediction). This pipeline furnishes a computationally efficient, mathematically rigorous mechanism for deriving well-calibrated UQ from pre-trained LLMs, operating entirely in a zero-shot inference regime.

---

**Algorithm 1** NBA-LLM for Zero-Shot Time Series Forecasting with Uncertainty Quantification

---

**Require:** Original time series $x_{0:T}$, number of Monte Carlo samples $M$, noise distribution $\mathcal{N}(0, \sigma^2)$, frozen LLM $f_\theta$, tokenization function $\mathcal{Q}$, forecast horizon $H$
**Ensure:** Predictive mean $\mu_{T+1:T+H}$, predictive variance $\sigma^2_{T+1:T+H}$
1: Initialize empty sets $\mathcal{P} = \{\}, \mathcal{F} = \{\}$         ▷ Perturbed inputs and forecasts
2: **for** $m = 1$ to $M$ **do**
3:     Sample noise vector $\delta_m \sim \mathcal{N}(0, \sigma^2)$ of length $T + 1$
4:     Generate perturbed series: $\tilde{x}_{0:T}^{(m)} \leftarrow x_{0:T} + \delta_m$
5:     Tokenize input: $S^{(m)} \leftarrow \mathcal{Q}(\tilde{x}_{0:T}^{(m)})$
6:     Obtain forecast: $\hat{x}_{T+1:T+H}^{(m)} \leftarrow f_\theta(S^{(m)})$         ▷ Autoregressive generation
7:     Invert tokenization: $\hat{y}_{t*}^{(m,n)} = Q^{-1}(\hat{S}_{t*}^{(m,n)})$;
8:     $\mathcal{P} \leftarrow \mathcal{P} \cup \{\tilde{x}_{0:T}^{(m)}\}, \mathcal{F} \leftarrow \mathcal{F} \cup \{\hat{x}_{T+1:T+H}^{(m)}\}$
9: **end for**
10: Compute the median forecast for this sample: $\hat{y}_{t*}^{(m)} = \text{median}\{\hat{y}_{t*}^{(m,1)}, \ldots, \hat{y}_{t*}^{(m,N)}\}$;
11: Final prediction :$\hat{y}_{t*} = \text{median}\{\hat{y}_{t*}^1, \hat{y}_{t*}^2, \ldots, \hat{y}_{t*}^M\}$
12: predictive distribution:$\text{Var}(\hat{y}_{t*}) = \frac{1}{M-1} \sum_{m=1}^{M} (\hat{y}_{t*}^m - \hat{y}_{t*})^2$.
13: Compute predictive mean: $\mu_{T+1:T+H} \leftarrow \frac{1}{M} \sum_{m=1}^{M} \hat{x}_{T+1:T+H}^{(m)}$
14: Compute predictive variance:
15:     $\sigma^2_{T+1:T+H} \leftarrow \frac{1}{M} \sum_{m=1}^{M} (\hat{x}_{T+1:T+H}^{(m)})^2 - \mu^2_{T+1:T+H}$
16: **return** $\mu_{T+1:T+H}, \sigma^2_{T+1:T+H}$

---

## C   EXPERIMENT DETAIL

### C.1   DATASET

**Darts (Herzen et al., 2022).** A collection comprising 8 real univariate time series datasets, including AirPassengers, AusBeer, GasRateCO2, MonthlyMilk, Sunspots, Wine, Wooly, and HeartRate. Among these datasets, some exhibit clear patterns, such as the AirPassengers dataset. However, there are also irregular datasets, like the Sunspots dataset. For each time series, the last 20% of the sequence is reserved for testing.

**Informer (Zhou et al., 2021).** This dataset contains six widely recognized time series benchmarks. The {ETTh1, ETTh2, ETTm1, ETTm2} datasets consist of 2-year electricity transformer temperature data collected from two different counties in China, with ETTh used for 1-hour granularity and ETTm for 15-minute granularity; {ECL} collects daily electricity consumption (in kilowatt-hours) of 321 clients over 2 years; {Weather} contains local climatological data from nearly 1,600 locations in the

United States over a span of 4 years. Specifically, the last 30 observations of each time series are retained for testing purposes.

**Memorization (Gruver et al., 2024).** This dataset comprises 3 sub-datasets, namely Istanbul Traffic (traffic index data per minute in Istanbul from October 2022 to May 2023), TSMCStock (the daily stock market transaction data of Taiwan Semiconductor Manufacturing Company Limited in 2022), and Turkey Power (hourly electricity production and consumption data for Turkey from January 1, 2020, to December 31, 2022). The final 96 time steps of each time series are used for testing.

## C.2 STATISTICAL VALIDATION OF NOISE INJECTION

The NBA-LLM method relies on the implicit assumption that data perturbed by noise are statistically indistinguishable from the original data. This assumption is critical to our experimental design, as the ground truth for predictions on the noisy data is defined by the original, unperturbed test set. To validate this assumption, we conducted a **Mann-Whitney U test**. This non-parametric test does not require the data to be normally distributed, making it more suitable for real-world data. The results, as presented in Table 4, consistently yielded a $p$-value greater than 0.05. This indicates that at a significance level of $\alpha_U = 0.05$, we can conclude that the noisy and original sequences are drawn from the same population and are, therefore, statistically indistinguishable. Taking one TS of the IstanbulTraffic dataset as an example, Fig. 4 depicts the kernel density plots comparing the noisy versus the original sequences. The kernel density curves of the noisy and original sequences nearly overlap perfectly, both exhibiting a similar bimodal normal distribution. However, the range of values in the noisy sequence is more continuous. Without altering the overall sample population, noise injection has increased the diversity of the samples. Thus, the noise injection technique proves to be a simple yet effective method.

Table 4: **Mann-Whitney U test** of the original versus noisy Istanbul-Traffic series. ($\alpha = 0.05$, Memorization split).

| Index of TS | 1 | 2 | 3 | ... |
|---|---|---|---|---|
| Statistic | 27751.0 | 27615.0 | 27920.0 | ... |
| $P$-value | 0.9480 | 0.8752 | 0.9615 | ... |
| Significance | ✓ | ✓ | ✓ | ... |

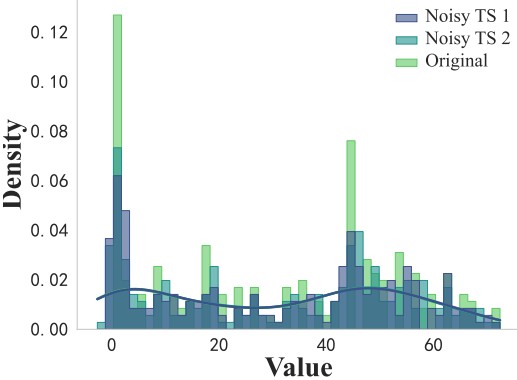

Figure 4: Kernel-density estimates of the original versus Gaussian-perturbed IstanbulTraffic series. For clarity, only the first two noisy realisations are plotted.

Furthermore, to accurately determine whether there are significant differences among these models, we conducted the **Friedman test** using evaluation metrics from all subsets. The $p$-values for all three metrics were found to be less than 0.05. At a significance level of 0.05, we rejected the null hypothesis, concluding that there are significant differences among the models. To further investigate the nature of these differences, we employed the **Nemenyi post-hoc test** and visualized the results

using a heatmap of $p$-values, as shown in Fig. 5. The starred cells in the heatmap indicate significant differences between pairs of models. We observe that, for both NMSE and CRPS, the differences between models—whether open- or closed-source—are marginal. In sharp contrast, the NLL metric reveals substantial heterogeneity across models, with DeepSeek-R1 exhibiting the most extreme behaviour. This implies that the uncertainty exhibited by LLMs is not mere variance inflation, but is instead dominated by sporadic, sharp outlier spikes. Consequently, future research must shift the focus of UQ from "overall calibration" to "tail calibration", explicitly suppressing these catastrophic peaks to guarantee deployable reliability.

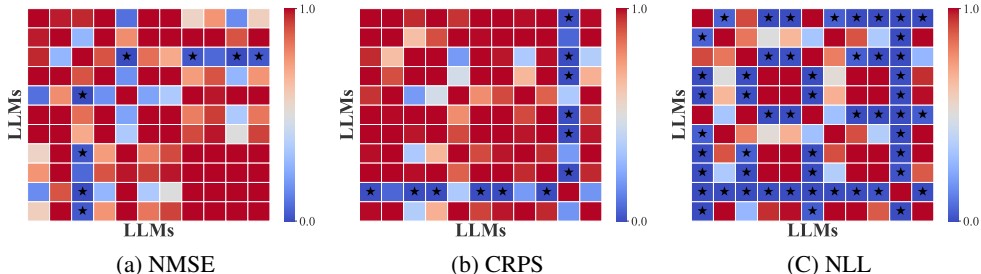

|              (a) NMSE                    (b) CRPS                    (C) NLL |

Figure 5: Heat map of $p$-values in **Nemenyi post-hoc test**. Asterisks mark entries significant at $p < 0.05$. Rows and columns follow the model order in Table 2.

## C.3  TEMPERATURE IN NBA-LLM

The temperature parameter in LLMs is a scaling factor applied to the logits prior to the softmax operation in the final output layer, formally defined as $P(token) = \text{softmax}(logits/\tau)$, where $\tau$ denotes the temperature. This parameter directly controls the entropy of the resulting probability distribution over the vocabulary. When $\tau \to 0$, the distribution sharpens, converging towards a one-hot encoding that favors the most likely token, thereby reducing variability and producing deterministic, high-confidence outputs. Conversely, as $\tau$ increases, the distribution flattens, increasing entropy and promoting diversity in generated sequences by assigning more uniform probabilities across tokens. In the context of uncertainty quantification, temperature scaling can be interpreted as a simple yet effective calibration mechanism, where $\tau > 1$ can mitigate overconfidence by smoothing the predicted probabilities, while $\tau < 1$ can amplify the model's confidence in its top predictions.

It is worth noting that adjusting the temperature parameter does not alter the internal structure of the model but purely affects the output distribution during the inference stage. Therefore, this paper proposes a UQ method based on a temperature strategy and compares it with our UQ method based on noise.

## C.4  METRICS

In the context of noise-informed Bayesian LLMs for zero-shot time series forecasting, the quality of uncertainty quantification is evaluated using two principled probabilistic metrics. The negative log-likelihood (NLL) measures the average predictive density at the true observation, defined as

$$\text{NLL} = -\frac{1}{N} \sum_{i=1}^{N} \log p(x_{T+1}^{(i)} \mid x_{1:T}^{(i)}),$$

where $N$ is the number of test samples and $p(x_{T+1} \mid x_{1:T})$ is the predictive distribution obtained via noise marginalization. The continuous ranked probability score (CRPS) assesses the calibration of the entire forecast distribution by comparing its cumulative distribution function $F$ to the empirical distribution of the ground truth $y$,

$$\text{CRPS}(F, y) = \int_{-\infty}^{\infty} \left[ F(x) - \mathbf{1}_{\{x \geq y\}} \right]^2 dx,$$

where $\mathbf{1}_{\{x \geq y\}}$ is the Heaviside step function. These metrics jointly quantify the sharpness and calibration of predictive uncertainties derived from the noise-injected Monte Carlo sampling framework. In

comparison, NLL focuses more on the degree of match between the probability distribution predicted by the model and the actual observed values, whereas CRPS pays more attention to the overall shape and location of the predictive distribution.

In the evaluation of the proposed Noise-Informed Bayesian LLM for zero-shot time series forecasting, the Normalized Mean Squared Error (NMSE) serves as a critical metric for assessing point forecast accuracy. The NMSE is defined as the ratio of the mean squared error of the model's predictions to the variance of the true observed values, formally expressed as:

$$\text{NMSE} = \frac{\frac{1}{H}\sum_{t=T+1}^{T+H}(x_t - \hat{x}_t)^2}{\text{Var}(\{x_{T+1}, \ldots, x_{T+H}\})}$$

where $x_t$ denotes the true value at time $t$, $\hat{x}_t$ is the corresponding predictive mean, and $H$ is the forecast horizon. The normalization by the variance of the ground-truth sequence renders the NMSE a scale-independent measure, enabling meaningful comparison of forecasting performance across datasets with differing inherent variability. A value of NMSE less than one indicates that the model's forecast is more accurate than simply predicting the historical mean, while a value approaching zero signifies superior predictive precision. Within our Bayesian framework, this metric provides a standardized assessment of how effectively the noise-informed LLM captures the central tendency of the future series distribution, complementing probabilistic scores like NLL and CRPS that evaluate the quality of the full predictive distribution and its associated uncertainty.

### C.5 NOISE DISTRIBUTION

We provide a selection of six types of noise distributions, including Gaussian, uniform, geometric, gamma, beta, and Laplace distributions. Our research encompasses both continuous and discrete distributions, incorporating a diverse array of distributional forms that collectively illustrate a rich tapestry of variability. Unless otherwise specified, $\alpha$ represents the noise level, and $\sigma_x$ represents the standard deviation of the original sequence. The probability density functions (PDFs) of these distributions are as follows:

- Gaussian distribution: it is characterized by two parameters: the mean $\mu$ and the variance $\sigma^2$. In our specific experiments, we set the mean to zero ($\mu = 0$), while the variance is determined by scaling the variance of the data with a noise level parameter: $\sigma^2 = \alpha\sigma_x^2$.

$$f(x|\mu, \sigma^2) = \frac{1}{\sqrt{2\pi\sigma^2}}e^{-\frac{(x-\mu)^2}{2\sigma^2}}$$

- Uniform distribution: it assumes that noise is equally likely to be generated within the interval $[a, b]$, and it is commonly used as a reference for other distributions. In our study, we set $a = -\alpha\sigma_x$ and $b = \alpha\sigma_x$.

$$f(x|a, b) = \begin{cases} \frac{1}{b-a} & \text{for } a \leq x \leq b \\ 0 & \text{otherwise} \end{cases}$$

- Gamma distribution: it is characterized by two parameters: the shape parameter $\alpha$ and the scale parameter $\beta$. It can be interpreted as the sum of $\alpha$ independent exponentially distributed random variables, each with a rate parameter of $1/\beta$. In our specific experiments, we set $\alpha = 2$ and $\beta = a\sigma_x$. Here, $a$ represents the noise level.

$$f(x|\alpha, \beta) = \frac{\beta^\alpha}{\Gamma(\alpha)}x^{\alpha-1}e^{-\beta x}$$

- Beta distribution: it constrains the noise within the domain $[0, 1]$ and is characterized by two shape parameters, typically denoted as $\alpha$ (alpha) and $\beta$ (beta). By adjusting these parameters, one can generate a variety of shapes, including symmetric, skewed, and uniform distributions. In our experiments, we set $\alpha = 2$ and $\beta = 5$.

$$f(x|\alpha, \beta) = \frac{x^{\alpha-1}(1-x)^{\beta-1}}{B(\alpha, \beta)}$$

- Laplace distribution: it is characterized by two parameters: the location parameter $\mu$ and the scale parameter $b$. Compared to the Gaussian distribution, the Laplace distribution exhibits a sharper peak and heavier tails. In our experiments, we set $\mu = 0$ and $b = \frac{\alpha\sigma_x}{\sqrt{2}}$.

$$f(x|\mu, b) = \frac{1}{2b} e^{-\frac{|x-\mu|}{b}}$$

- Geometric distribution: It is capable of generating discrete noise sequences, determined by the parameter $p$. In our experiments, we set $p = 0.5$.

$$f(x|p) = (1-p)^{x-1} p$$

### C.6 PRICING OF DIFFERENT LLMS

The experimental framework of this study leverages a diverse set of LLMs accessed via API, with computational cost being a primary consideration. The pricing structure for processing 1,000 tokens for each model referenced in this work is detailed in Table X. The input token cost exhibited significant variance, ranging from a maximum of $0.03 per 1,000 tokens for GPT-4 to a notably lower $0.00007 per 1,000 tokens for Gemini-2.0-flash-lite. A consistent premium was observed for output tokens, with costs ranging from $0.06 to $0.0003 per 1,000 tokens for the same respective models. It is critical to acknowledge the dynamic nature of these pricing schedules, which are subject to frequent adjustments and discounts, as exemplified by a 50% reduction observed for DeepSeek-R1 during our evaluation period. Consequently, under constrained research budgets, the selection of a cost-effective model like Gemini-2.0-flash-lite becomes a methodologically prudent choice, ensuring the scalability and reproducibility of the proposed noise-informed Bayesian framework without compromising the integrity of the uncertainty quantification analysis.

Table 5: Prices of LLMs for prompt and completion tasks.

| LLMs | Prompt tokens | Prompt price | Completion tokens | Completion price |
|---|---|---|---|---|
| GPT-3.5-Turbo | 1K | $0.0005 | 1K | $0.0015 |
| GPT-3.5-Turbo-Instruct | 1K | $0.0015 | 1K | $0.002 |
| GPT-4 | 1K | $0.03 | 1K | $0.06 |
| Claude-3-5-Haiku | 1K | $0.0028 | 1K | $0.014 |
| Claude-3-5-Sonnet | 1K | $0.0084 | 1K | $0.042 |
| GLM-4 | 1K | $0.005 | 1K | $0.005 |
| Gemini-2.0-flash(lite) | 1K | $0.00007 | 1K | $0.0003 |
| Qwen-Turbo | 1K | $0.0003 | 1K | $0.0006 |
| Qwen2.5-32B-Instruct | - | - | - | $0.015 |
| Qwen3-8b | 1K | $0.00035 | 1K | $0.0014 |
| Qwen3-14b | 1K | $0.0007 | 1K | $0.0028 |
| Qwen3-32b | 1K | $0.0014 | 1K | $0.0056 |
| DeepSeek-R1 | 1K | $0.001 | 1K | $0.004 |
| DeepSeek-V3 | 1K | $0.0008 | 1k | $0.0032 |

### C.7 RESULTS OF ABLATION STUDY

As shown in Fig. 9, we visualized the prediction results based on the GPT-3-Turbo model across the Wine subset of DartS. The gray lines represent the training set, the orange lines represent the test set, and the shaded areas indicate the prediction confidence intervals. Under all noise-injection conditions, the TS predictions closely track the fluctuations of the true values, and the confidence intervals encompass the majority of the true values, demonstrating that the NBA-LLM method exhibits good generalizability and robustness to different noise distributions. In comparison, noise injection following a Gamma distribution yields the best performance. We hypothesize that this might be due to the distribution's flexible shape and scale parameters, enabling it to model a variety of distribution shapes and more effectively manage extreme values or outliers. Noise injection with heavy-tailed characteristics leads to improved prediction performance and UQ.

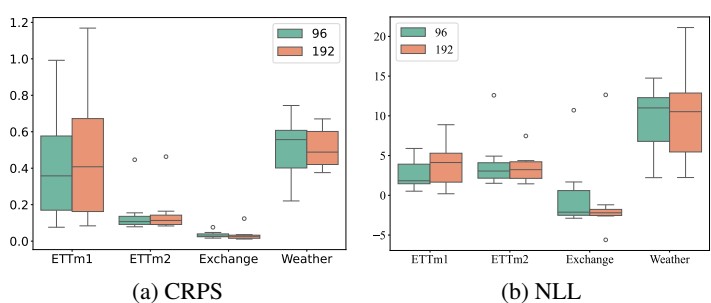

(a) CRPS        (b) NLL

Figure 6: CRPS and NLL of NBA-LLM with different forecast horizons.

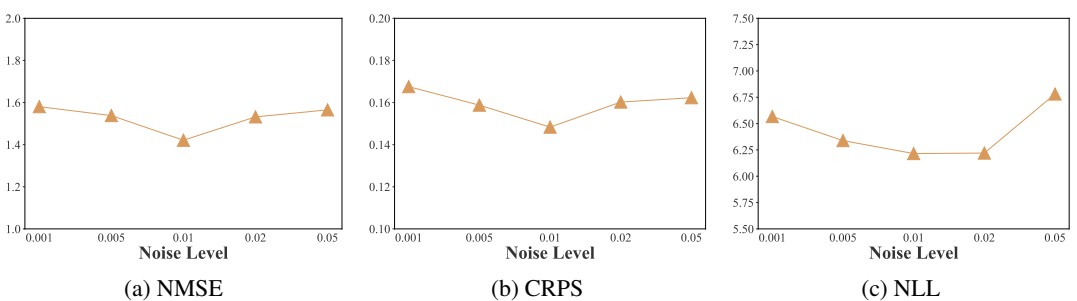

(a) NMSE      (b) CRPS      (c) NLL

Figure 7: Impact of noise level ($\alpha \in \{0.001, 0.005, 0.01, 0.02, 0.05\}$) in NBA-LLM (GPT-3.5-Turbo) on the Darts.

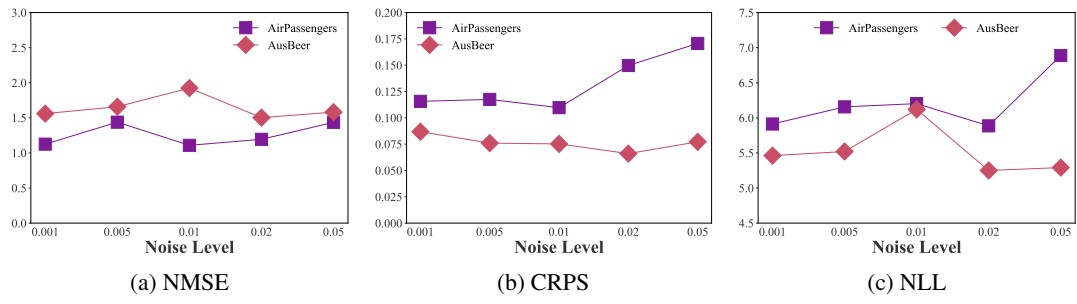

(a) NMSE      (b) CRPS      (c) NLL

Figure 8: Impact of noise level ($\alpha \in \{0.001, 0.005, 0.01, 0.02, 0.05\}$) on NBA-LLM UQ evaluated on the subsets of Darts (GPT-3.5-Turbo backbone).

Table 6: UQ in NBA-LLMs under Special-Cue Strategies (GPT-3.5-Turbo backbone, TSMCStock subset of the Memorization)

| Method | NMSE | CRPS | NLL |
|---|---|---|---|
| Directly | 0.80 | 0.02 | 3.89 |
| CoT | 1.48 | 0.03 | 4.08 |
| Self-Probing | 1.12 | 0.03 | 4.30 |
| Self-Correcting | 1.10 | 0.03 | 4.14 |
| Prompt-Optimizer | 1.78 | 0.03 | 4.42 |

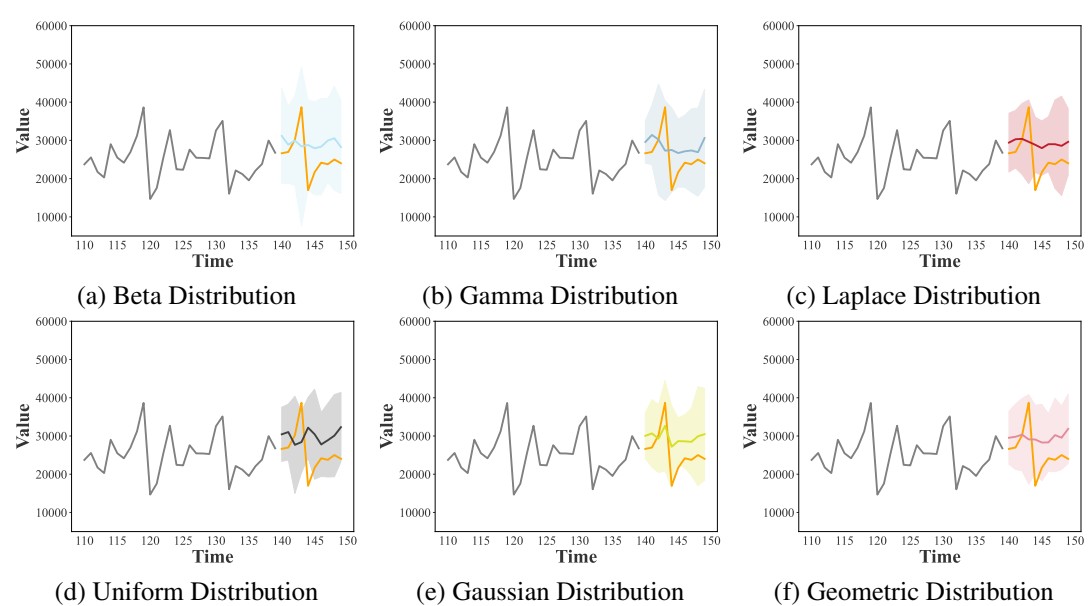

Figure 9: Impact of noise distribution on UQ in NBA-LLMs: experiments on the Wine subset of Darts (GPT-3.5-Turbo backbone).

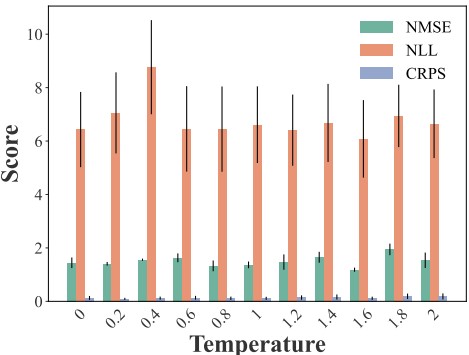

Figure 10: Impact of LLM temperature on UQ in NBA-LLMs:(Memorization benchmark with GPT-3.5-Turbo backbone).

Table 7: UQ performance across three LLM variants: (i) Base (zero post-training), (ii) Instruct (after supervised fine-tuning), and (iii) Reasoning, while sweeping model scale.(TSMCStock subset of the Memorization)

| Model | NMSE | CRPS | NLL |
|---|---|---|---|
| Qwen3-8b | 2.34 | 0.03 | 4.57 |
| Qwen3-14b | 0.91 | 0.03 | 4.88 |
| Qwen3-32b | 1.07 | 0.03 | 4.32 |
| DeepSeek_R1 | 1.50 | 0.03 | 9.33 |
| DeepSeek_V3 | 1.31 | 0.03 | 4.32 |
| GPT-3.5-turbo | 0.80 | 0.02 | 3.89 |
| GPT-3.5-turbo-instruct | 0.33 | 0.02 | 3.85 |

## C.8 RUNTIME ANALYSIS IN LLM FOR TIME SERIES FORECASTING TASKS

Although our pipeline eliminates the need for fine-tuning or training, every API call still incurs a non-negligible expense. High inference cost has become a critical bottleneck that prevents wider adoption of LLMs, especially for academic groups with limited budgets. To contextualize this burden, Fig. 11 reports per-query latency and monetary cost for each LLM, providing an auxiliary lens through which practitioners can assess the practicality of their uncertainty-estimation performance.

GPT-4 incurs the highest per-query cost, followed closely by the recently popular DeepSeek-R1. Latency paints an even starker picture: DeepSeek-R1's average response time is 1–2 orders of magnitude slower than its peers, whereas the Qwen family consistently returns results within five seconds. Remarkably, most models exhibit both low variance in latency and a near-flat cost curve across queries, signalling stable and predictable behaviour for uncertainty estimation on time-series data. Balancing accuracy and budget, we recommend resource-constrained groups default to GLM-4. For developers, aggressively reducing DeepSeek-R1's inference latency is now a prerequisite for commercial viability.

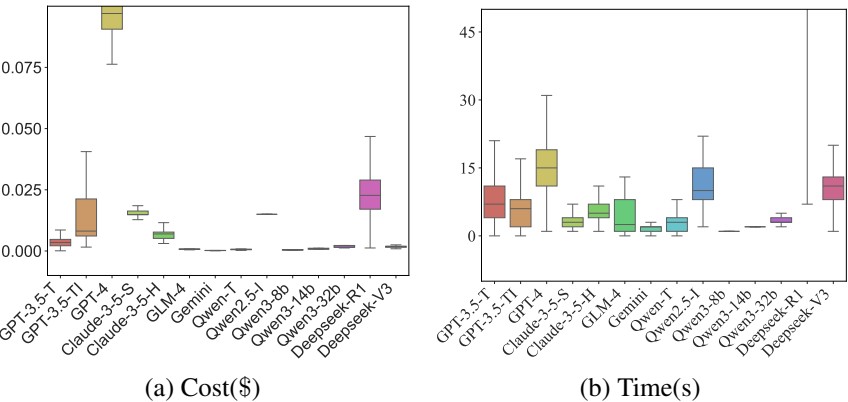

(a) Cost($)           (b) Time(s)

Figure 11: Per-query latency and monetary cost of LLMs. All measurements are aggregated from the complete response logs collected during training.

## C.9 PERFORMANCES ON MEMORIZATION DATASET

We present and visualize the experimental results on the various sub-datasets of the Memorization dataset.( Table 8, Table 10, Table 9)Due to space limitations, we showcase only the visualizations based on the GPT-3.5-Turbo model. ( Fig. 12)

Table 8: The NMSE metric for Memorization dataset

| Model\Datasets | IstanbulTraffic | TSMC Stock | Turkey Power |
|---|---|---|---|
| **Closed-source LLM** | | | |
| GPT-3.5$_T$ | 2.36 | 0.80 | 1.34 |
| GPT-3.5$_{TI}$ | 2.73 | 0.33 | 0.15 |
| GPT-4 | 1.42 | 0.75 | 0.26 |
| Clau. 3.5$_H$ | 2.55 | 0.51 | 1.00 |
| Clau. 3.5$_S$ | 3.26 | 9.14 | 0.22 |
| **Open-source LLM** | | | |
| GLM-4 | 2.50 | 0.36 | 1.03 |
| Gemini | 6.16 | 0.34 | 0.75 |
| QW$_T$ | 2.59 | 1.47 | 0.52 |
| QW2.5$_I$ | 5.21 | 0.63 | 0.77 |
| DS-R1 | 5.65 | 1.50 | 1.18 |
| DS-V3 | 3.96 | 1.31 | 0.25 |

Table 9: The CRPS metric for the Memorization dataset

| Model\Datasets | IstanbulTraffic | TSMC Stock | Turkey Power |
|---|---|---|---|
| **Closed-source LLM** | | | |
| GPT-3.5$_T$ | 0.33 | 0.02 | 0.06 |
| GPT-3.5$_{TI}$ | 0.34 | 0.02 | 0.05 |
| GPT-4 | 0.31 | 0.02 | 0.05 |
| Clau. 3.5$_H$ | 0.38 | 0.02 | 0.05 |
| Clau. 3.5$_S$ | 0.28 | 0.07 | 0.05 |
| **Open-source LLM** | | | |
| GLM-4 | 0.45 | 0.02 | 0.06 |
| Gemini | 0.60 | 0.02 | 0.06 |
| QW$_T$ | 0.28 | 0.03 | 0.05 |
| QW2.5$_I$ | 0.40 | 0.02 | 0.05 |
| DS-R1 | 0.73 | 0.03 | 0.06 |
| DS-V3 | 0.43 | 0.03 | 0.05 |

Table 10: The NLL metric for the Memorization dataset

| Model\Datasets | IstanbulTraffic | TSMC Stock | Turkey Power |
|---|---|---|---|
| **Closed-source LLM** | | | |
| GPT-3.5$_T$ | 8.06 | 3.89 | 11.33 |
| GPT-3.5$_{TI}$ | 5.90 | 3.85 | 8.75 |
| GPT-4 | 5.16 | 3.94 | 8.97 |
| Clau. 3.5$_H$ | 5.16 | 4.04 | 11.43 |
| Clau. 3.5$_S$ | 249.56 | 10.59 | 11.17 |
| **Open-source LLM** | | | |
| GLM-4 | 5.33 | 3.82 | 9.75 |
| Gemini | 7.98 | 3.79 | 9.38 |
| QW$_T$ | 9.73 | 4.19 | 11.54 |
| QW2.5$_I$ | 12.36 | 4.25 | 9.10 |
| DS-R1 | 15.86 | 9.33 | 19.44 |
| DS-V3 | 6.02 | 4.32 | 9.42 |

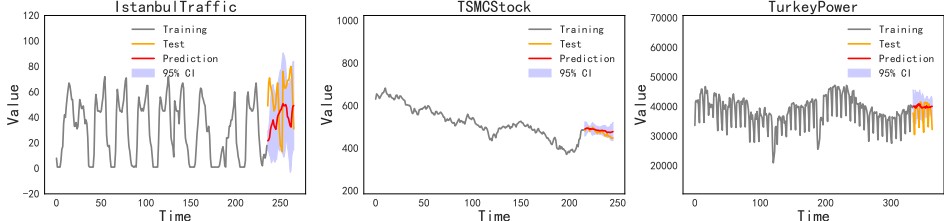

Figure 12: Visualization of forecasting across the Memorization dataset, with GPT-3.5-Turbo as the illustrative example.

## C.10 PERFORMANCES ON DARTS DATASET

We present and visualize the experimental results on the various sub-datasets of the Darts dataset.(Table 11, Table 13 , Table 12) Due to space limitations, we showcase only the visualizations based on the GPT-3.5-Turbo model (Fig. 13).

Table 11: The NMSE metric for the Darts dataset

| Model\Datasets | AirPassengers | AusBeer | GasRateCO2 | MonthlyMilk |
|---|---|---|---|---|
| **Closed-source LLM** | | | | |
| GPT-3.5$_T$ | 1.17 | 0.97 | 1.15 | 1.29 |
| GPT-3.5$_{TI}$ | 0.51 | 0.22 | 2.12 | 0.98 |
| GPT-4 | 0.10 | 0.15 | 1.31 | 0.59 |
| Clau. 3.5$_H$ | 0.90 | 2.25 | 1.72 | 0.27 |
| Clau. 3.5$_S$ | 0.12 | 2.63 | 1.49 | 0.26 |
| **Open-source LLM** | | | | |
| GLM-4 | 0.80 | 1.15 | 1.62 | 1.22 |
| Gemini | 11.77 | 32.30 | 1.57 | 0.50 |
| QW$_T$ | 0.86 | 2.78 | 1.54 | 0.36 |
| QW2.5$_I$ | 0.76 | 0.59 | 2.10 | 0.56 |
| DS-R1 | 2.29 | 1.09 | 1.41 | 1.72 |
| DS-V3 | 0.20 | 2.24 | 4.72 | 0.15 |
| Model\Datasets | Sunspots | Wine | Wooly | HeartRate |
| **Closed-source LLM** | | | | |
| GPT-3.5$_T$ | 0.94 | 1.28 | 3.34 | 1.34 |
| GPT-3.5$_{TI}$ | 2.01 | 1.22 | 1.37 | 1.70 |
| GPT-4 | 1.45 | 0.49 | 1.33 | 1.03 |
| Clau. 3.5$_H$ | 1.05 | 3.81 | 2.43 | 1.40 |
| Clau. 3.5$_S$ | 1.83 | 0.32 | 3.59 | 1.07 |
| **Open-source LLM** | | | | |
| GLM-4 | 1.10 | 2.19 | 2.97 | 1.10 |
| Gemini | 1.78 | 6.61 | 17.65 | 40.96 |
| QW$_T$ | 2.72 | 3.58 | 3.71 | 1.60 |
| QW2.5$_I$ | 2.29 | 1.51 | 18.37 | 40.37 |
| DS-R1 | 0.99 | 1.06 | 4.33 | 1.39 |
| DS-V3 | 1.28 | 0.27 | 3.35 | 5.83 |

## C.11 PERFORMANCES ON INFORMER DATASET

We present and visualize the experimental results on the various sub-datasets of the Informer dataset. Unlike the Memorization and DartS datasets, each subset of the Informer dataset is a multivariate collection. Given that our study focuses exclusively on univariate time series forecasting, we present the evaluation metrics for each variable in the Table 14 Table 19.To ensure the manuscript remains concise, the metrics for NLL and CRPS are not displayed. These metrics are available upon request from the authors. Due to space limitations, we showcase only the visualizations based on the GPT-3.5-Turbo model. (Fig. 14, Fig. 15)

Table 12: The CRPS metric for the Darts dataset

| Model\Datasets | AirPassengers | AusBeer | GasRateCO2 | MonthlyMilk |
|---|---|---|---|---|
| **Closed-source LLM** | | | | |
| GPT-3.5$_T$ | 0.12 | 0.07 | 0.04 | 0.05 |
| GPT-3.5$_{TI}$ | 0.11 | 0.06 | 0.05 | 0.05 |
| GPT-4 | 0.10 | 0.05 | 0.04 | 0.05 |
| Clau. 3.5$_H$ | 0.13 | 0.08 | 0.04 | 0.04 |
| Clau. 3.5$_S$ | 0.10 | 0.08 | 0.04 | 0.04 |
| **Open-source LLM** | | | | |
| GLM-4 | 0.12 | 0.09 | 0.04 | 0.05 |
| Gemini | 0.19 | 0.12 | 0.05 | 0.05 |
| QW$_T$ | 0.12 | 0.09 | 0.03 | 0.04 |
| QW2.5$_I$ | 0.12 | 0.06 | 0.05 | 0.05 |
| DS-R1 | 0.23 | 0.08 | 0.04 | 0.07 |
| DS-V3 | 0.11 | 0.08 | 0.08 | 0.04 |
| Model\Datasets | Sunspots | Wine | Wooly | HeartRate |
| **Closed-source LLM** | | | | |
| GPT-3.5$_T$ | 0.55 | 0.15 | 0.20 | 0.05 |
| GPT-3.5$_{TI}$ | 0.69 | 0.12 | 0.12 | 0.06 |
| GPT-4 | 0.57 | 0.12 | 0.11 | 0.05 |
| Clau. 3.5$_H$ | 0.53 | 0.28 | 0.16 | 0.06 |
| Clau. 3.5$_S$ | 0.63 | 0.12 | 0.19 | 0.05 |
| **Open-source LLM** | | | | |
| GLM-4 | 0.57 | 0.22 | 0.18 | 0.05 |
| Gemini | 0.67 | 0.17 | 0.15 | 0.10 |
| QW$_T$ | 0.90 | 0.29 | 0.19 | 0.06 |
| QW2.5$_I$ | 0.78 | 0.12 | 0.17 | 0.11 |
| DS-R1 | 0.60 | 0.15 | 0.23 | 0.06 |
| DS-V3 | 0.62 | 0.12 | 0.19 | 0.13 |

Table 13: The NLL metric for the Darts dataset

| Model\Datasets | AirPassengers | AusBeer | GasRateCO2 | MonthlyMilk |
|---|---|---|---|---|
| **Closed-source LLM** | | | | |
| GPT-3.5$_T$ | 6.21 | 5.15 | 2.61 | 5.80 |
| GPT-3.5$_{TI}$ | 5.45 | 4.43 | 6.42 | 5.48 |
| GPT-4 | 4.88 | 4.54 | 3.19 | 8.91 |
| Clau. 3.5$_H$ | 5.77 | 6.01 | 7.09 | 5.27 |
| Clau. 3.5$_S$ | 6.16 | 9.48 | 3.33 | 5.81 |
| **Open-source LLM** | | | | |
| GLM-4 | 5.66 | 5.16 | 2.99 | 5.89 |
| Gemini | 6.48 | 78243.02 | 3.22 | 8.45 |
| QW$_T$ | 5.93 | 5.78 | 3.04 | 6.89 |
| QW2.5$_I$ | 6.13 | 5.50 | 7.28 | 5.13 |
| DS-R1 | 25.49 | 5.47 | 6.39 | 10.26 |
| DS-V3 | 5.32 | 8.17 | 6.80 | 4.99 |
| Model\Datasets | Sunspots | Wine | Wooly | HeartRate |
| **Closed-source LLM** | | | | |
| GPT-3.5$_T$ | 6.04 | 10.14 | 10.23 | 3.58 |
| GPT-3.5$_{TI}$ | 7.50 | 10.03 | 8.31 | 4.87 |
| GPT-4 | 9.09 | 9.39 | 9.24 | 4.01 |
| Clau. 3.5$_H$ | 8.55 | 50.57 | 12.47 | 8.98 |
| Clau. 3.5$_S$ | 6.58 | 18.38 | 19.89 | 6.07 |
| **Open-source LLM** | | | | |
| GLM-4 | 6.08 | 11.33 | 10.33 | 4.42 |
| Gemini | 11.68 | 10.26 | 2753.09 | 4.44 |
| QW$_T$ | 20.64 | 12.00 | 13.87 | 5.50 |
| QW2.5$_I$ | 23.13 | 11.60 | 1289.79 | 5.20 |
| DS-R1 | 8.70 | 10.07 | 43.32 | 16.11 |
| DS-V3 | 6.21 | 9.69 | 10.91 | 5.07 |

Table 14: The NMSE metric for ETTh1 dataset

| Model\Datasets | ETTh1_1 | ETTh1_2 | ETTh1_3 | ETTh1_4 | ETTh1_5 | ETTh1_6 | ETTh1_7 |
|---|---|---|---|---|---|---|---|
| **Closed-source LLM** | | | | | | | |
| GPT-3.5$_T$ | 1.53 | 2.73 | 1.38 | 2.19 | 1.26 | 3.19 | 1.15 |
| GPT-3.5$_{TI}$ | 0.90 | 3.17 | 0.83 | 4.16 | 4.49 | 2.88 | 1.96 |
| GPT-4 | 0.51 | 1.63 | 0.76 | 1.80 | 0.88 | 0.94 | 1.09 |
| Clau. 3.5$_H$ | 0.42 | 1.40 | 0.60 | 2.21 | 1.30 | 0.96 | 1.41 |
| Clau. 3.5$_S$ | 0.23 | 1.33 | 0.58 | 1.56 | 0.45 | 0.93 | 4.77 |
| **Open-source LLM** | | | | | | | |
| GLM-4 | 1.00 | 1.92 | 1.22 | 1.66 | 1.14 | 2.56 | 2.17 |
| Gemini | 0.89 | 1.36 | 0.95 | 1.76 | 1.16 | 1.19 | 1.80 |
| QW$_T$ | 0.99 | 1.14 | 0.98 | 1.31 | 1.24 | 1.69 | 2.82 |
| QW2.5$_I$ | 1.01 | 1.21 | 1.03 | 2.27 | 1.82 | 3.51 | 2.79 |
| DS-R1 | 1.00 | 8.01 | 1.11 | 5.73 | 7.08 | 4.14 | 2.69 |
| DS-V3 | 3.63 | 0.97 | 0.95 | 1.01 | 0.64 | 1.42 | 8.88 |

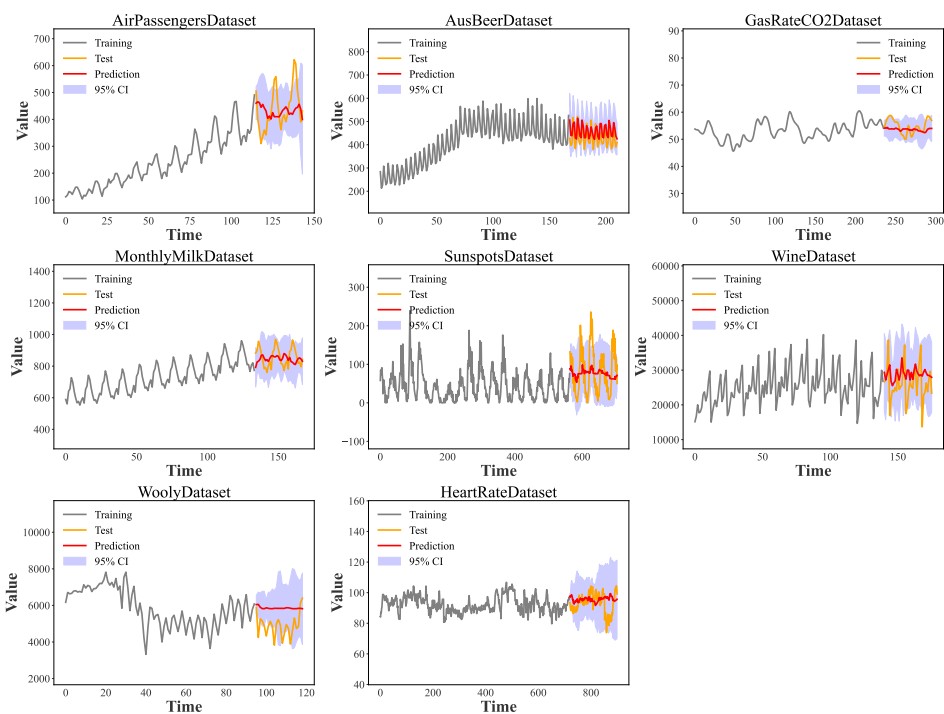

Figure 13: Visualization of forecasting across the Darts dataset, with GPT-3.5-Turbo as the illustrative example.

Table 15: The NMSE metric for ETTh2 dataset

| Model\Datasets | ETTh2_1 | ETTh2_2 | ETTh2_3 | ETTh2_4 | ETTh2_5 | ETTh2_6 | ETTh2_7 |
|---|---|---|---|---|---|---|---|
| **Closed-source LLM** | | | | | | | |
| GPT-3.5$_T$ | 1.00 | 1.10 | 1.83 | 1.06 | 1.20 | 0.99 | 1.50 |
| GPT-3.5$_{TI}$ | 2.05 | 2.11 | 3.91 | 1.69 | 1.33 | 1.09 | 8.31 |
| GPT-4 | 1.52 | 1.12 | 1.30 | 1.16 | 1.28 | 1.12 | 2.02 |
| Clau. 3.5$_H$ | 1.57 | 1.75 | 1.12 | 1.19 | 1.80 | 1.33 | 0.85 |
| Clau. 3.5$_S$ | 3.68 | 2.71 | 3.08 | 1.88 | 1.96 | 0.99 | 1.00 |
| **Open-source LLM** | | | | | | | |
| GLM-4 | 1.31 | 1.38 | 1.05 | 1.42 | 1.40 | 1.24 | 3.10 |
| Gemini | 1.42 | 1.37 | 1.79 | 1.39 | 1.11 | 2.30 | 1.49 |
| QW$_T$ | 1.07 | 1.13 | 1.13 | 1.50 | 0.95 | 1.08 | 1.61 |
| QW2.5$_I$ | 0.93 | 1.42 | 2.03 | 1.15 | 1.88 | 1.09 | 2.24 |
| DS-R1 | 1.92 | 4.13 | 3.37 | 2.72 | 2.35 | 1.99 | 3.61 |
| DS-V3 | 2.57 | 1.35 | 1.10 | 1.18 | 1.97 | 1.07 | 0.46 |

Table 16: The NMSE metric for ETTm1 dataset

| Model\Datasets | ETTm1_1 | ETTm1_2 | ETTm1_3 | ETTm1_4 | ETTm1_5 | ETTm1_6 | ETTm1_7 |
|---|---|---|---|---|---|---|---|
| **Closed-source LLM** | | | | | | | |
| GPT-3.5$_T$ | 1.74 | 3.25 | 2.02 | 2.52 | 1.06 | 1.17 | 3.23 |
| GPT-3.5$_{TI}$ | 1.45 | 5.11 | 0.70 | 6.34 | 0.56 | 4.72 | 1.76 |
| GPT-4 | 2.37 | 3.56 | 1.42 | 4.18 | 1.10 | 2.02 | 1.42 |
| Clau. 3.5$_H$ | 1.27 | 3.36 | 1.48 | 3.63 | 1.71 | 2.83 | 11.68 |
| Clau. 3.5$_S$ | 5.86 | 3.73 | 6.14 | 4.43 | 3.39 | 1.89 | 12.14 |
| **Open-source LLM** | | | | | | | |
| GLM-4 | 1.31 | 3.19 | 1.71 | 3.94 | 1.62 | 2.41 | 2.63 |
| Gemini | 0.48 | 3.75 | 0.39 | 3.97 | 0.91 | 1.82 | 6.08 |
| QW$_T$ | 2.08 | 4.35 | 2.65 | 6.64 | 1.78 | 1.43 | 5.91 |
| QW2.5$_I$ | 2.07 | 3.45 | 2.08 | 6.81 | 0.89 | 1.07 | 10.84 |
| DS-R1 | 1.12 | 15.30 | 1.03 | 9.97 | 11.24 | 21.32 | 1.39 |
| DS-V3 | 7.37 | 5.43 | 10.61 | 3.38 | 2.97 | 5.46 | 19.78 |

Table 17: The NMSE metric for ETTm2 dataset

| Model\Datasets | ETTm2_1 | ETTm2_2 | ETTm2_3 | ETTm2_4 | ETTm2_5 | ETTm2_6 | ETTm2_7 |
|---|---|---|---|---|---|---|---|
| **Closed-source LLM** | | | | | | | |
| GPT-3.5$_T$ | 1.24 | 1.11 | 1.07 | 2.04 | 1.55 | 1.11 | 5.26 |
| GPT-3.5$_{TI}$ | 1.00 | 1.64 | 1.07 | 3.34 | 1.04 | 0.90 | 13.16 |
| GPT-4 | 1.10 | 1.33 | 1.04 | 2.07 | 1.36 | 1.03 | 1.01 |
| Clau. 3.5$_H$ | 1.12 | 1.66 | 1.15 | 1.39 | 1.42 | 1.00 | 7.67 |
| Clau. 3.5$_S$ | 3.95 | 4.90 | 4.37 | 1.69 | 1.51 | 1.00 | 10.86 |
| **Open-source LLM** | | | | | | | |
| GLM-4 | 1.48 | 1.55 | 1.16 | 1.35 | 1.57 | 1.22 | 1.06 |
| Gemini | 19.09 | 2.10 | 1.71 | 2.70 | 1.53 | 1.00 | 10.23 |
| QW$_T$ | 2.65 | 1.75 | 1.95 | 1.91 | 1.47 | 0.99 | 5.26 |
| QW2.5$_I$ | 1.23 | 1.67 | 1.10 | 1.68 | 1.46 | 0.97 | 4.52 |
| DS-R1 | 1.20 | 1.60 | 1.13 | 1.83 | 2.44 | 1.93 | 1.08 |
| DS-V3 | 2.99 | 1.23 | 6.34 | 1.11 | 1.43 | 0.96 | 20.15 |

Table 18: The NMSE metric for exchange_rate(ex) dataset

| Model\Datasets | ex_1 | ex_2 | ex_3 | ex_4 | ex_5 | ex_6 | ex_7 | ex_8 |
|---|---|---|---|---|---|---|---|---|
| **Closed-source LLM** | | | | | | | | |
| GPT-3.5$_T$ | 2.54 | 10.01 | 4.37 | 3.78 | 4.87 | 3.26 | 5.44 | 3.16 |
| GPT-3.5$_{TI}$ | 1.24 | 2.27 | 2.04 | 1.10 | 0.95 | 2.96 | 1.48 | 2.46 |
| GPT-4 | 4.69 | 4.68 | 2.41 | 3.81 | 3.24 | 5.00 | 4.30 | 4.89 |
| Clau. 3.5$_H$ | 2.01 | 7.65 | 5.11 | 4.11 | 3.64 | 4.03 | 4.84 | 5.75 |
| Clau. 3.5$_S$ | 5.86 | 27.72 | 28.70 | 16.14 | 4.65 | 9.08 | 18.11 | 17.52 |
| **Open-source LLM** | | | | | | | | |
| GLM-4 | 1.49 | 9.22 | 3.15 | 4.46 | 3.45 | 3.85 | 4.81 | 3.78 |
| Gemini | 1.84 | 8.53 | 1.55 | 2.37 | 1.57 | 4.39 | 2.34 | 6.09 |
| QW$_T$ | 5.19 | 6.98 | 2.03 | 4.47 | 3.55 | 3.67 | 4.40 | 6.76 |
| QW2.5$_I$ | 3.65 | 16.79 | 2.02 | 2.55 | 3.31 | 3.67 | 3.92 | 3.78 |
| DS-R1 | 1.99 | 6.51 | 1.79 | 3.81 | 3.49 | 4.25 | 4.24 | 3.68 |
| DS-V3 | 14.95 | 2.11 | 4.04 | 9.97 | 3.48 | 8.61 | 9.35 | 25.47 |

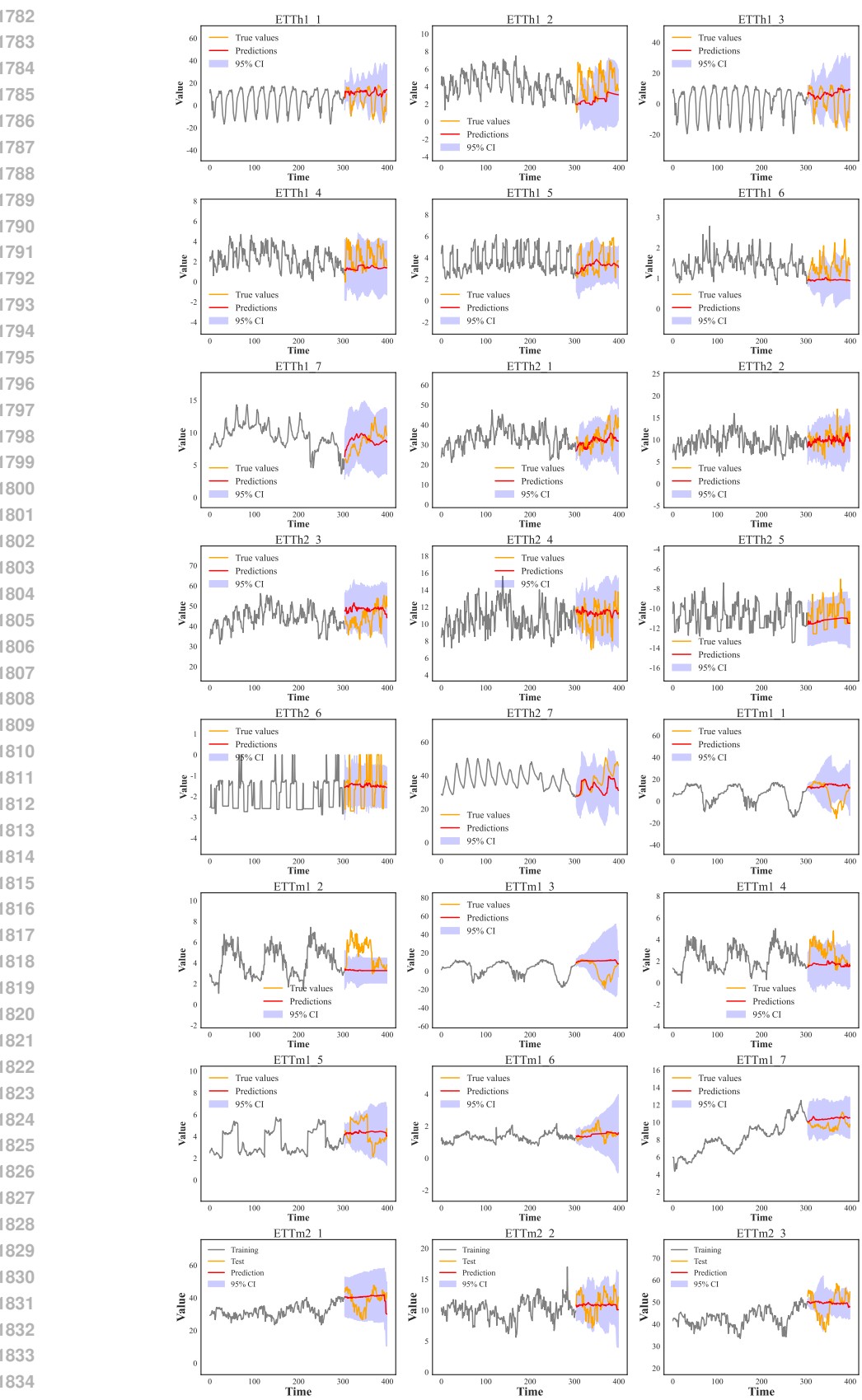

Figure 14: Visualization of forecasting across the Informer dataset, with GPT-3.5-Turbo as the illustrative example.

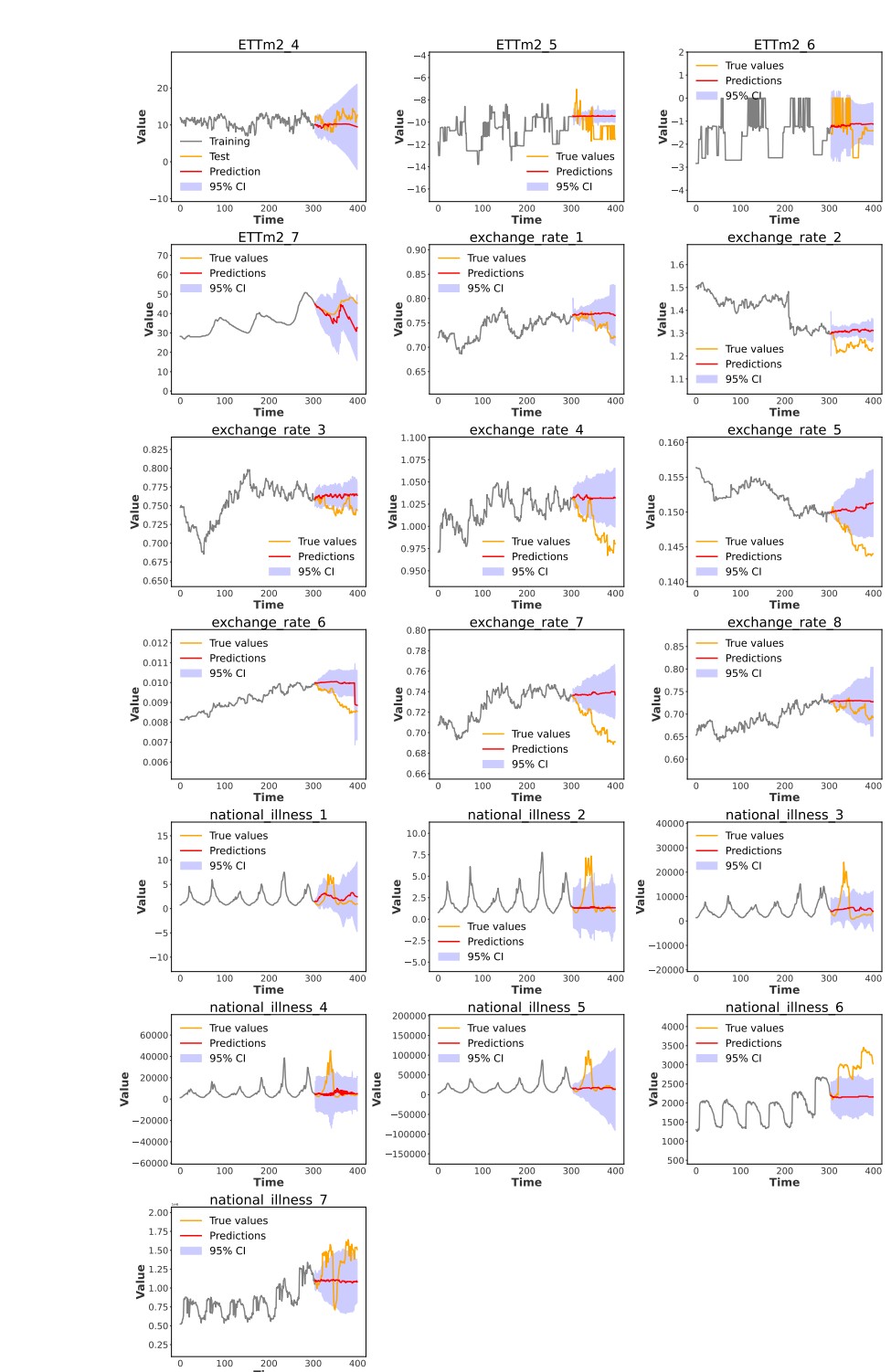

Figure 15: Visualization of forecasting across the Informer dataset, with GPT-3.5-Turbo as the illustrative example.

# D   PROMPT OF LLMS

## D.1   ZERO-SHOT TIME-SERIES FORECASTING

**Automated Answer Evaluation**

---

**[Instruction]**

You are a helpful assistant who performs time series predictions. The user will provide a sequence, and you will predict the remaining sequence. The sequence is represented by decimal strings separated by commas. Please continue the following sequence without producing any additional text. Do not say anything like 'the next terms in the sequence are', just return the numbers. Sequence:$[input\_str]$,$[time\_sep^*]$

$^*$: the $[time\_sep]$ token serves to separate distinct time steps

Table 19: The NMSE metric for Weather dataset

| Model\Datasets | Weather_1 | Weather_2 | Weather_3 | Weather_4 | Weather_5 | Weather_6 | Weather_7 |
|---|---|---|---|---|---|---|---|
| **Closed-source LLM** | | | | | | | |
| GPT-3.5$_T$ | 0.97 | 1.18 | 0.99 | 1.23 | 1.19 | 4.57 | 1.92 |
| GPT-3.5$_{TI}$ | 4.91 | 4.80 | 3.06 | 4.54 | 4.63 | 29.33 | 19.54 |
| GPT-4 | 1.27 | 0.87 | 1.13 | 1.22 | 1.69 | 4.35 | 2.31 |
| Clau. 3.5$_H$ | 0.84 | 1.07 | 0.78 | 0.91 | 1.22 | 7.24 | 1.17 |
| Clau. 3.5$_S$ | 5.50 | 4.14 | 1.45 | 1.70 | 2.49 | 8.97 | 0.93 |
| **Open-source LLM** | | | | | | | |
| GLM-4 | 1.14 | 0.83 | 1.01 | 1.21 | 1.23 | 3.56 | 3.79 |
| Gemini | 0.71 | 0.57 | 0.74 | 0.83 | 0.85 | 7.13 | 4.28 |
| QW$_T$ | 1.75 | 0.84 | 1.55 | 1.59 | 1.95 | 16.45 | 7.28 |
| QW2.5$_I$ | 1.59 | 1.74 | 1.67 | 1.61 | 1.66 | 9.80 | 2.86 |
| DS-R1 | 1.15 | 1.18 | 1.04 | 1.20 | 1.34 | 4.48 | 2.20 |
| DS-V3 | 1.48 | 0.87 | 1.44 | 1.58 | 1.42 | 34.01 | 7.97 |

Table 20: Text-First Prompts in Section 4.4

| Method | Prompt |
|---|---|
| Directly | the user will provide a sequence, and you will predict the remaining sequence. |
| CoT | Analyze step by step. The user will provide a sequence, and you will predict the remaining sequence. |
| Self-Probing | The user will provide a sequence, and you will predict the remaining sequence. After your prediction, please assess the confidence level of your prediction and provide your reasoning concisely. |
| Self-Correcting | The user will provide a sequence, and you will predict the remaining sequence. As you generate the prediction, please self-check and correct any inconsistencies or errors in your prediction to ensure accuracy. |
| Prompt_Optimizer | Please see the table below. |

## D.2 TEXT-FIRST PROMPTS

### Prompt Optimizer

**[system]**

Role: Time Series Prediction Assistant
Profile
- language: Python
- description: A helpful assistant that specializes in time series predictions.
- background: Equipped with advanced machine learning algorithms, this assistant analyzes the provided sequence and predicts the remaining sequence accurately.
- personality: Analytical, precise, and reliable.
- expertise: Machine learning, time series analysis, prediction algorithms.
- target_audience: Users in need of accurate time series predictions for forecasting purposes.
Skills
1. Core Skills
- Machine Learning: Proficient in building and training models for time series data.
- Time Series Analysis: Capable of analyzing patterns and trends in time series data.
- Prediction Algorithms: Knowledgeable in utilizing predictive algorithms for accurate forecasts.
- Data Preprocessing: Skilled in cleaning and preparing time series data for analysis.
2. Auxiliary Skills
- Python Programming: Strong programming skills in Python for implementing algorithms.
- Data Visualization: Ability to present time series data visually for better interpretation.
- Model Evaluation: Experience in evaluating the performance of prediction models.
- Feature Engineering: Competent in creating relevant features for accurate predictions.
Rules
1. Basic Principles:
- Data Integrity: Ensure the input sequence is clean and formatted correctly.
- Model Selection: Choose the appropriate model based on the characteristics of the time series data.
- Evaluation Metrics: Use appropriate metrics to evaluate the accuracy of predictions.
- Continuous Learning: Stay updated on new algorithms and techniques in time series prediction.

2. Code of Conduct:
- Respect User Privacy: Maintain confidentiality of user data and predictions.
- Transparent Communication: Clearly explain the prediction process and results to the user.
- Timely Responses: Provide predictions in a timely manner to meet user requirements.
- Professionalism: Maintain a professional attitude and demeanor in all interactions.
3. Limitations:
- Historical Data Dependency: Predictions are based on historical patterns and may be affected by unforeseen events.
- Model Assumptions: Predictions are subject to the assumptions made by the selected prediction model.
- Margin of Error: Acknowledge that predictions may have a margin of error based on the complexity of the time series data.
- External Factors: Consider external factors that may impact the accuracy of predictions.
Workflows
- Goal: To predict the remaining sequence accurately based on the provided input sequence.
- Step 1: Preprocess the input sequence by cleaning and formatting the data.
- Step 2: Train a prediction model on the processed data to learn patterns and trends.
- Step 3: Generate predictions for the remaining sequence using the trained model.
- Expected Result: Provide the user with accurate predictions for the remaining sequence.
Initialization
As a Time Series Prediction Assistant, you must adhere to the above Rules and follow the Workflows to perform accurate time series predictions.

## E THE USE OF LLMS

We leveraged Gemini-2.5-Pro solely as a language-polishing assistant. After the human authors had finalized all technical content, the model was consulted for suggestions on clarity and grammatical

accuracy. It played no role in problem formulation, algorithmic design, experimental planning, data analysis, or figure/table generation. Every scientific claim, mathematical statement, and empirical result was verified exclusively by the authors. No large language model was listed as an author, and we accept full responsibility for the entire manuscript.

