# OpenReview forum: "Noise Informed LLM for Zero-shot Time Series Forecasting with Uncertainty Quantification"
_ICLR.cc/2026/Conference — ICLR 2026 Conference Withdrawn Submission_

### Official Review · Reviewer_VNso · 2025-10-20

**Soundness:** 3
**Presentation:** 2
**Contribution:** 2
**Rating:** 2
**Confidence:** 3

**Summary:**

The paper addresses the lack of uncertainty quantification in LLM-based time series forecasting by proposing a model-agnostic noise-informed Bayesian approximation (NBA) framework that quantifies the uncertainty of frozen LLMs.

**Strengths:**

- The proposed approach is model-agnostic and works with frozen LLMs for zero-shot prediction capabilities.
- The authors release anonymized open-source code to promote reproducibility.
- Table 1 offers a clear overview of existing LLM-based methods for uncertainty quantification.
- The use of CRPS appropriately evaluates distributional forecasts and reflects attention to predictive uncertainty.
- The ablation studies, including examining the effects of noise and forecast horizon, provide some useful insights.

**Weaknesses:**

1. While the paper demonstrates a solid understanding of Large Language Models (LLMs), it shows limited awareness of fundamental concepts, methodologies, and recent advances in the time series forecasting (TSF) domain, despite time series being a primary focus suggested by the title.
   - For example, the authors claim that (lines 38-40), “By leveraging intricate prompt engineering and tokenization mechanisms (Naveed et al., 2024), the application of LLMs to TS forecasting represents an emerging and surprisingly effective paradigm, capitalizing on their innate ability to discern and extrapolate complex temporal patterns in a zero-shot manner.”  However, this assertion is debatable. Several studies have reported mixed or negative evidence regarding the effectiveness of general-purpose LLMs for time series forecasting. This gap in performance has motivated the development of an entire class of Time Series Foundation Models (TSFMs) specifically designed for such tasks.
    - The statement that “A significant limitation of this approach lies in its predominant focus on deterministic point predictions, neglecting a cornerstone of trustworthy forecasting: the quantification of predictive uncertainty.” overlooks a major body of work in probabilistic forecasting, where predictive uncertainty is explicitly modeled through probabilistic loss functions (e.g., CRPS, quantile losses). Many TSFMs already incorporate such approaches. A more grounded discussion within the broader TS forecasting context would strengthen the paper.
    - Regarding lines 154–155: the paper suggests that time series forecasting generally involves decomposing data into structural signal and stochastic noise components. This is not a general property of time series forecasting. While some architectures, such as DLinear, N-BEATS, and Autoformer, aim to extract trend, seasonality, and/or noise components, many modern transformer-based or deep learning models do not rely on explicit decomposition.
    - The authors also claim that (lines 157-159), “The objective of TS forecasting extends beyond point prediction to the probabilistic estimation of future values {xT +1, xT +2, . . . , xT +H } over a horizon H, conditioned on historical observations.” While probabilistic forecasting is indeed an important and growing research direction, it is not the universal objective of TS forecasting. Rather, it represents a recent paradigm shift aimed at improving uncertainty quantification.

2.  The prior work referenced for Uncertainty Quantification (UQ) pertains mainly to text data. The paper does not justify why this approach is not evaluated on text data as well.

3.  The paper does not reference to recent LLM-based forecasting models such as Time-LLM, Chronos, and other LLM-based forecasting frameworks.

4. The UQ methods listed in Table 1 are not consistently evaluated in Table 2, which limits the ability to assess comparative performance.

5. It is unclear what exactly Table 2 demonstrates regarding the proposed UQ technique.  It would be more informative to present forecasting results with and without the proposed UQ method, to isolate its contribution. Similarly, in the forecasting figures, no baseline model is shown, making it difficult to assess relative performance.

**Questions:**

1. Lines 262–263: How are time series tokenized for the LLM in this work? Is the tokenization based on discrete binning, quantization, or another encoding scheme?

---

### Official Review · Reviewer_zoD5 · 2025-10-30

**Soundness:** 2
**Presentation:** 3
**Contribution:** 1
**Rating:** 2
**Confidence:** 4

**Summary:**

The paper introduces a method for zero-shot time-series forecasting using LLMs' in-context learning abilities. In particular, the authors propose to obtain uncertainty estimates for LLM predictions by injecting noise into the input time series and evaluating the mean / variance of predictions obtained for $M$ of the resulting noised time series. The authors evaluate this method over a small synthetic dataset sampled from a GP, as well as some standard time series datasets (Darts, Informer, Memorization). The authors also run ablations over different noise levels and try alternative prompting strategies.

**Strengths:**

- The idea of injecting noise into the time series to improve the calibration of LLMs is in itself interesting and would be worth exploring in more details.
- The results are evaluated over multiple LLMs.

**Weaknesses:**

- **Related works are not discussed or compared against**: the idea of using LLMs as zero-shot time series forecasters is not new, and yet the paper fails to discuss existing works in this area or compare (qualitatively or empirically) against the already established uncertainty evaluation methods for time series forecasting with LLMs. In particular, LLMTime [1] provides a comprehensive study of using LLMs for time-series forecasting, also comparing against state-of-the-art time series forecasting methods. LLMProcesses [2] provides a method for using the logits of the numerical tokens of the LLMs to obtain uncertainty scores. These works provide comprehensive evaluations of LLM's time series forecasting capabilities over multiple different datasets, classes of functions, time series serialisation and tokenisation schemes and uncertainty estimation methods. Further, to make the uncertainty quantification more computationally-efficient, [3] proposes to use a probing model to estimate the inherent LLM uncertainty directly from its hidden states. In other words, this area is already well-established and many of the conclusions and experiments of this paper feel obsolete.
- **The stochasticity of LLMs**: a natural way of estimating the uncertainty of LLMs is by using their inherent stochasticity. For example, one could generate 100 samples from the LLM and calculate the resulting mean and variance (as done by [1]). Alternatively, one can try to use the logits of each token to get a more detailed estimate of the LLM's distribution (as done by [2]). However, it is not clear to me (1) how (whether) this work utilises the stochasticity of LLMs (do the authors use greedy sampling? How many samples are generated for a single noised version of the time series?), (2) how the proposed NBA-LLM compares against these baseline methods of obtaining uncertainty of LLMs (does using noise rather than sampling lead to more well-calibrated results?)
- **Lack of comparisons**: Although the method compares the performance of multiple different LLMs, no non-LLM methods are provided as baselines which would allow to contextualise the obtained results and evaluate whether the proposed noising approach indeed leads to reasonable results and improves performance compared to alternative strategies. As an example, following [2], the authors could try comparing the obtained negative log likelihood against that of a Gaussian Process. For me, the most important experiment would be to see whether injecting noise into the input time series allows to better calibration of the LLM outputs than simply using LLM samples for a non-perturbed input time series, yet this experiment is also lacking.
- **Lack of details**: Finally, the method is not described in sufficient details in the paper, making it difficult to fully evaluate the validity of the results (see questions below).

[1] Gruver et al. 2023 (https://proceedings.neurips.cc/paper_files/paper/2023/hash/3eb7ca52e8207697361b2c0fb3926511-Abstract-Conference.html)
[2] Requeima et al. 2024 (https://proceedings.neurips.cc/paper_files/paper/2024/hash/c5ec22711f3a4a2f4a0a8ffd92167190-Abstract-Conference.html)
[3] Piskorz et al. 2025 (https://openreview.net/forum?id=6PPh2N7LZ2)

**Questions:**

- In this framework, how are the samples obtained from the LLM? Greedily? By sampling? What sampling method was used? What is the temperature?
- How are the time series values serialised?
- How many noise samples $M$ are used across experiments? What is the effect of the number of samples $M$ on the calibration?
- How is the negative log likelihood (NLL) computed in the experiments? Is the predictive distribution approximated as a Gaussian, using the estimated mean and variance?
- In the appendix, the authors show that performance is not monotone with the changing noise level. Is there a noise level which consistently improves calibration across all datasets? If not, how can the optimal noise be chosen?

---

### Official Review · Reviewer_QQ83 · 2025-10-30

**Soundness:** 1
**Presentation:** 2
**Contribution:** 1
**Rating:** 2
**Confidence:** 4

**Summary:**

The paper proposes to noise inputs into transformers to estimate uncertainty. This is applied to time-series forecasting tasks. The method is applied on 1D regression tasks.

**Strengths:**

1. Time-series forecasting with LLMs is an interesting topic
2. The appendix contains some detailed and interesting experiments

**Weaknesses:**

In order of magnitude

1. The paper exceeds the allowed 9 pages, and page 8 already uses suspiciously small vspaces.
2. The method is not novel: Inputs to an LLM are noised and the variance of the outputs is taken as an uncertainty estimate.
3. The method is never compared to any baselines, although UQ for LLMs has a rich field of Bayesian methods (GP, Laplace, Deep Ensembles, not even simple resampling of the model’s output distribution...). The only baseline, taking an integral over different temperature values in Table 3, is wrong by design and I have not seen it used in literature.
4. In the experiments that show interpretable comparisons, Figure 3 a and c, the test values are outside the 95% interval generated with the model. Figures 14 and 15 in the appendix suggest that this happens far more often than 95% of the time, so I would challenge the claim that the proposed method is calibrated.
5. The paper claims that the predicted variance is “provably calibrated to epistemic plus aleatoric uncertainty”. Epistemic and aleatoric uncertainty are never discussed, neither the provable calibration, and I indeed believe it cannot be calibrated (see Question 1).
6. The paper claims to be compute efficient (unlike other methods), but resampling a transformer multiple times is not compute efficient.
7. This paper argues from a very high level here that any sort of integral over some \theta, be it actual noise in the model parameters or any other form of noise such as input noise, is essentially Bayesian, and so their method which uses input noise is a Bayesian approximation. However I believe most Bayesians would challenge the idea that averaging over input perturbations is actually estimating a Bayesian posterior, because the interesting and hard challenge is modeling the uncertainty over the model parameters \theta
8. Much of the math in this paper is artificially complicated. E.g., lines 254-261 explain in great detail and notation what a tokenizer is. This notation is not required to explain the paper, and in fact not even used in the next paragraph: In Equation 7, S and Q (tokenizer and detokenizer) are not used anymore, instead it is now called Token(), and the derived equations are even wrong and not using the previous insight that tokenization should be applied forward and inversely backwards. The correct form would be p(x+1) = p(detokenize(z) | tokenize(x)), not p(x+1) = p(tokenize(z) | tokenize (x)).

### Smaller weaknesses that don’t influence my score and don’t need rebuttal but would be nice to consider in the final version

* Some of the text appears bloated with overcomplicated terms, or adjectives that are unnecessary. I would recommend to simplify these.
* The contribution of the paper might be more fitting for a workshop or short paper.
* Appendix C.6: Table X should be Table 5

### Justification for score

The paper proposes a simple, known mechanism, noising input variables, and applies it to simplistic 1D regression examples. It does not compare to any baselines, despite discussing them. There are claims that are not discussed by theory or experiments. I recommend to reject and resubmit an extended version to a workshop or short paper track.

**Questions:**

1. Why introduce input noise to the LLM? You could just resample the LLM multiple times to get an uncertainty. This uncertainty is even guaranteed to be calibrated (in the limit of large enough training data), whereas input-noised uncertainties are not guaranteed to represent the true data-generating process’es variance (only the variance of a flattened version of it, that's why you observe the additional term in your variance formula)
2. What is your input noise “calibrated”?

---

### Official Review · Reviewer_JeTo · 2025-11-01

**Soundness:** 2
**Presentation:** 3
**Contribution:** 2
**Rating:** 2
**Confidence:** 4

**Summary:**

This paper introduces a method to perform black-box uncertainty quantification (UQ) for LLM-based time-series forecasting by marginalizing over noised versions of the observed sequence, motivated by a Bayesian formulation. The paper presents experiments showing the predictive accuracy and uncertainty metrics across datasets and LLMs and visualization of the induced predictive uncertainty.

**Strengths:**

- The proposed method is simple and can be applied with any models.
- Empirically, the induced predictive uncertainty looks visually plausible.
- Experiments cover multiple commonly used datasets.
- The motivation and related works are well-presented.

**Weaknesses:**

- Despite the theoretical motivation, I'm afraid I find the method is far from principled. Marginalization involves integrating the predictive distribution over the posterior, $p(\delta|x_{1:T})$, using the paper's notation, but the authors never discuss how to compute $p(\delta|x_{1:T})$, and instead simply swaps it with some prior $p(\delta)$ they specify, e.g. zero-mean Gaussian. I did not find an explanation for how this can be justified. This is not Bayesian marginalization. A analogous move was made for the temperature temparameter in L409.
- With this change, the method corresponds to simply adding some arbitrarily defined noise to the input and averaging the LLM's predictive distribution. I do not see how this can be claimed as a principled UQ method.
- The authors use Proposition 4 for claim that they that "proved that a mathematically grounded model uncertainty estimate" can be obtained via noisy prompts. This proposition is simply the law of total variance, and holds for arbitrary choices of the noisy $p(\delta).$ As such I do not see how this result shows their UQ method is mathematically grounded.
- I do not see evidence that the proposed method improves calibration over out-of-the-box LLM, such as by NLL or CRPS.

**Questions:**

- Why is it ok to swap the posterior for $p(\delta|x_{1:T})$ and $p(\tau|x_{1:T})$ with the respective priors?
- How much does the method improve NLL or CRPS over out-of-the-box LLM?
- How did you obtain log probabilities from the tested LLMs? Prior work (e.g. Gruver et al.) reported log probs returned by the API from models such as GPT-3, but I believe newer models including GPT-4 and Claude 3.5 do not provide log probs.

---

### Note · Authors · 2025-11-20

I have read and agree with the venue's withdrawal policy on behalf of myself and my co-authors.